# Energy stress promotes P-bodies formation via lysine-63-linked polyubiquitination of HAX1

Wanqi Zhan[1,2,3,10], Zhiyang Li[1,2,3,10], Jie Zhang [4,5,10], Yongfeng Liu [6,10], Guanglong Liu[1,2,3], Bingsong Li [1,7], Rong Shen[1,2,3], Yi Jiang[4,5], Wanjing Shang[8], Shenjia Gao[4,5], Han Wu[4,5], Ya'nan Wang[1,2,3], Wankun Chen [4,5,9✉] & Zhizhang Wang [1,7✉]

## Abstract

**Energy stress, characterized by the reduction of intracellular ATP, has been implicated in various diseases, including cancer. Here, we show that energy stress promotes the formation of P-bodies in a ubiquitin-dependent manner. Upon ATP depletion, the E3 ubiquitin ligase TRIM23 catalyzes lysine-63 (K63)-linked polyubiquitination of HCLS1-associated protein X-1 (HAX1). HAX1 ubiquitination triggers its liquid–liquid phase separation (LLPS) and contributes to P-bodies assembly induced by energy stress. Ubiquitinated HAX1 also interacts with the essential P-body proteins, DDX6 and LSM14A, promoting their condensation. Moreover, we find that this TRIM23/HAX1 pathway is critical for the inhibition of global protein synthesis under energy stress conditions. Furthermore, high HAX1 ubiquitination, and increased cytoplasmic localization of TRIM23 along with elevated HAX1 levels, promotes colorectal cancer (CRC)-cell proliferation and correlates with poor prognosis in CRC patients. Our data not only elucidate a ubiquitination-dependent LLPS mechanism in RNP granules induced by energy stress but also propose a promising target for CRC therapy.**

**Keywords** Energy Stress; P-bodies; TRIM23; HAX1; Translation Inhibition
**Subject Categories** Cancer; RNA Biology

## Introduction

Adenosine triphosphate (ATP) is the most abundant and primary chemical energy required for various biological processes in normal cells. Energy stress, characterized by reduced intracellular ATP levels due to factors such as lessened blood flow, disturbed oxygen supply, or imbalanced energy metabolism, has been associated with numerous human diseases, including neurodegenerative disorders and cancer (Butterfield & Halliwell, 2019; Martinez-Outschoorn et al, 2017). Eukaryotic cells respond to energy stress through adaptive programs that include activation of AMP-activated protein kinase (AMPK) and inhibition of global translation to re-establish energy homeostasis and maintain cell survival (Jeon et al, 2012). However, the mechanisms underlying the translational restrictions imposed by energy stress remain less understood.

Ribonucleoprotein (RNP) granules are membraneless cellular compartments with microscopically visible features that govern various aspects of RNA metabolism (Schisa, 2014). One prominent subtype of RNP granule is the RNA processing body (P-body), a cytoplasmic superaggregate that is either constitutive in certain mammalian cell lines or induced in response to stressors (Kedersha et al, 2005). P-bodies are enriched with mRNAs that are translationally repressed, along with protein factors related to mRNA metabolism. This composition enables rapid cellular adaptation to environmental stress by modulating gene expression at post-transcriptional levels, without affecting transcription (Eulalio et al, 2007). While ATP is essential for P-bodies disassembly (Mugler et al, 2016), the connection between energy state and P-bodies formation has yet to be established.

The liquid-like puncta of P-bodies condense through a process known as multicomponent liquid-liquid phase separation (LLPS) involving proteins and RNAs (Brangwynne et al, 2009). This process is mediated by several types of molecular interactions, including 'classical' high-affinity interactions of RNA binding proteins (RBPs) (Ferraiuolo et al, 2005; Kamenska et al, 2016), low-affinity interactions between low-complexity sequence domains (LCDs) (Kato et al, 2012) or intrinsically disordered regions (IDRs) (Nott et al, 2015), and multivalent interactions between proteins and RNAs (Teixeira et al, 2005). However, the specific molecular mechanisms underlying the assembly of P-bodies through LLPS are still obscure. Several RBPs, including the decapping enzyme DCP1A (Lykke-Andersen, 2002; van Dijk et al, 2002), the decapping activator EDC4 (Yu et al, 2005), the eIF4E-binding protein 4E-T (Nishimura et al, 2015), the RNA helicase DDX6

[1]Department of Pathology, Nanfang Hospital, Southern Medical University, Guangzhou, Guangdong, China. [2]Department of Pathology, School of Basic Medical Sciences, Southern Medical University, Guangzhou, Guangdong, China. [3]Guangdong Province Key Laboratory of Molecular Tumor Pathology, Guangzhou, Guangdong, China. [4]Department of Anesthesiology, Zhongshan Hospital, Fudan University, Shanghai, China. [5]Shanghai Key Laboratory of Perioperative Stress and Protection, Shanghai, China. [6]Radiation Medicine Institute, The First Affiliated Hospital, ZhengZhou University, ZhengZhou, Henan, China. [7]Jinfeng Laboratory, Chongqing, China. [8]Lymphocyte Biology Section, Laboratory of Immune System Biology, National Institute of Allergy and Infectious Diseases, National Institutes of Health, Bethesda, USA. [9]Department of Anesthesiology, Qingpu Branch of Zhongshan Hospital, Fudan University, Shanghai, China. [10]These authors contributed equally: Wanqi Zhan, Zhiyang Li, Jie Zhang, Yongfeng Liu. ✉E-mail: chen.wankun@zs-hospital.sh.cn; wangzz89@smu.edu.cn

(Coller et al, 2001), LSm family proteins LSm1-7 (Ingelfinger et al, 2002) and LSM14A (Yang et al, 2006), the 5′ to 3′ exonuclease XRN1 (Bashkirov et al, 1997), and the HCLS1-associated protein X-1 (HAX1) (Zayat et al, 2015), have been identified as components of P-bodies, whereas most proteins have been proven as unnecessary for P-bodies formation (Aizer et al, 2008; Huch et al, 2016). LSM14A and DDX6 are increasingly recognized as requisites for P-bodies assembly (Ayache et al, 2015; Hubstenberger et al, 2017), but the mechanisms by which they are recruited to the P-bodies under specific stress conditions have not been determined.

An increase in ubiquitination is another hallmark of cellular response to stress, which conventionally activates the selective proteolysis of stress-induced damaged proteins or regulates protein function in multiple ways (Franzmann and Alberti, 2021). Ubiquitin signaling-associated proteins, such as the E3 ligases TRAF6 and TRIM24, significantly influence P-bodies dynamics by ubiquitinating multiple P-bodies proteins (Tenekeci et al, 2016; Wei et al, 2022). In this study, we find that energy stress promotes P-bodies formation in a ubiquitin-dependent manner through the TRIM23-HAX1 pathway. Specifically, the E3 ligase TRIM23 links the K63 polyubiquitin chain to HAX1 when ATP is depleted. Ubiquitinated HAX1 undergoes LLPS, promoting the condensation of LSM14A and DDX6 during P-bodies formation. Moreover, we reveal that the TRIM23/HAX1 pathway is critical for the inhibition of protein synthesis under energy stress. Finally, we discover that the TRIM23/HAX1 pathway is critical for the tumorigenicity of CRC, suggesting a potential mechanism by which cancer cells adapt to low ATP conditions to promote tumorigenesis.

# Results

## Energy stress promotes P-bodies formation and dynamics

To investigate the potential impact of energy stress on P-bodies assembly, we depleted intracellular ATP by blocking the glycolytic pathway and oxidative phosphorylation using 2-deoxyglucose (2DG) and oligomycin, respectively. This intervention resulted in a substantial reduction in cellular ATP levels ranging from 30% to 80% within 4 h (Fig. EV1A). A low concentration of intracellular ATP suppressed sodium arsenite-induced stress granules formation (Fig. EV1B,C), consistent with previous studies showing that stress granules assembly requires ATP (Jain et al, 2016). Surprisingly, the addition of 2DG and/or oligomycin boosted the formation of P-bodies under basal conditions in HEK293T cells (Fig. 1A,B), and oligomycin treatment increased the number of P-bodies in RKO cells (Fig. EV1D,E). The P-bodies induced by energy stress exhibited a heterogeneous size distribution (Fig. 1A), and the number of both large and small P-bodies increased under energy stress (Fig. EV1F). Despite oligomycin alone reduced cellular ATP levels by 50% compared to the 80% reduction observed with 2-DG plus oligomycin, it was sufficient to increase the number of P-bodies from one to four per HEK293T cell, a response comparable to that induced by the combination of 2-DG plus oligomycin. We chose oligomycin alone for ATP depletion in subsequent experiments.

Conversely, addition of coenzyme Q10 (CoQ10) and reduced nicotinamide adenine dinucleotide (NADH) (Castro-Marrero et al, 2021; Lenaz et al, 1997), known to augment cellular ATP production via mitochondrial oxidative phosphorylation (Fig. EV1G), hindered P-bodies formation under both basal and heat shock conditions in HET293 cells (Fig. 1C,D) and RKO cells (Fig. EV1H,I). Increasing intracellular ATP levels using ATPsome (Fig. EV1J) slightly reduced the number of P-bodies under basal conditions but remarkably prevented P-bodies formation induced by heat shock (Figs. 1E,F and EV1K). Moreover, P-bodies formation induced by ATP depletion was reversed upon supplementation with ATPsome in HEK293 cells (Figs. 1G,H and EV1L) and RKO cells (Fig. EV1M,N). We next examined whether energy stress regulates P-bodies dynamics by fluorescence recovery after photobleaching (FRAP) assay. As expected, oligomycin treatment significantly enhanced the total recovery of LSM14A-GFP (Fig. 1I,J). These results suggest that energy stress activates P-bodies formation in an ATP-dependent manner.

## Energy stress induces protein ubiquitination to favor P-bodies formation

Protein ubiquitination increases when eukaryotic cells respond to different stresses (Franzmann and Alberti, 2021), and the ubiquitin signaling pathway has been proven to regulate P-bodies assembly (Kedia et al, 2022). Thus, we hypothesized that energy stress promoted P-bodies formation by regulating protein ubiquitination. To test this, we first confirmed that ubiquitinated proteins accumulated after ATP depletion (Figs. 2A and EV2A), which could be partially reversed with ATPsome supplementation (Fig. 2B). To further investigate whether ubiquitylation is essential to P-bodies formation induced by energy stress, we treated HEK293T cells with two E1-activating enzyme (UAE) inhibitors, TAK-243 and PYR-41 (Hyer et al, 2018; Yang et al, 2007), to disrupt the ubiquitylation process (Fig. EV2B). Acute pharmacological UAE inhibition resulted in a significant reduction in P-bodies number under both basal and oligomycin-stimulated conditions (Figs. 2C,D and EV2C,D). This inhibitory effect was also observed in HCT116 cells treated with TAK-243 and PYR-41 (Fig. EV2E–G). Interestingly, proteasome inhibition by MG132 and RA190 had little effect on P-bodies formation (Figs. 2E,F and EV2H,I), suggesting the involvement of a nondegradative ubiquitylation pathway in mediating P-bodies formation. Similarly, knocking down ubiquitin decreased the number of P-bodies in HEK293T cells under different stress conditions (Figs. 2G,H and EV2J–L). Conversely, the overexpression of ubiquitin induced the formation of P-bodies, which primarily depended on lysine-63 (K63) ubiquitin linkages (Figs. 2I,J and EV2M,N), consistent with findings from previous studies (Hao et al, 2013).

Moreover, TAK-243 and PYR-41 treatment reduced the total recovery of LSM14A-GFP, with PYR-41 treatment significantly slowing down the FRAP process (Fig. 2K,L). Knocking down ubiquitin almost completely abolished the recovery ability of LSM14A after photobleaching (Fig. 2M,N). In contrast, forced ubiquitin expression facilitated the recovery of LSM14A (Fig. EV2O,P), suggesting that ubiquitination is necessary for P-bodies dynamics. Taken together, these data indicate that energy stress regulates P-bodies dynamics via K63-linked protein ubiquitination.

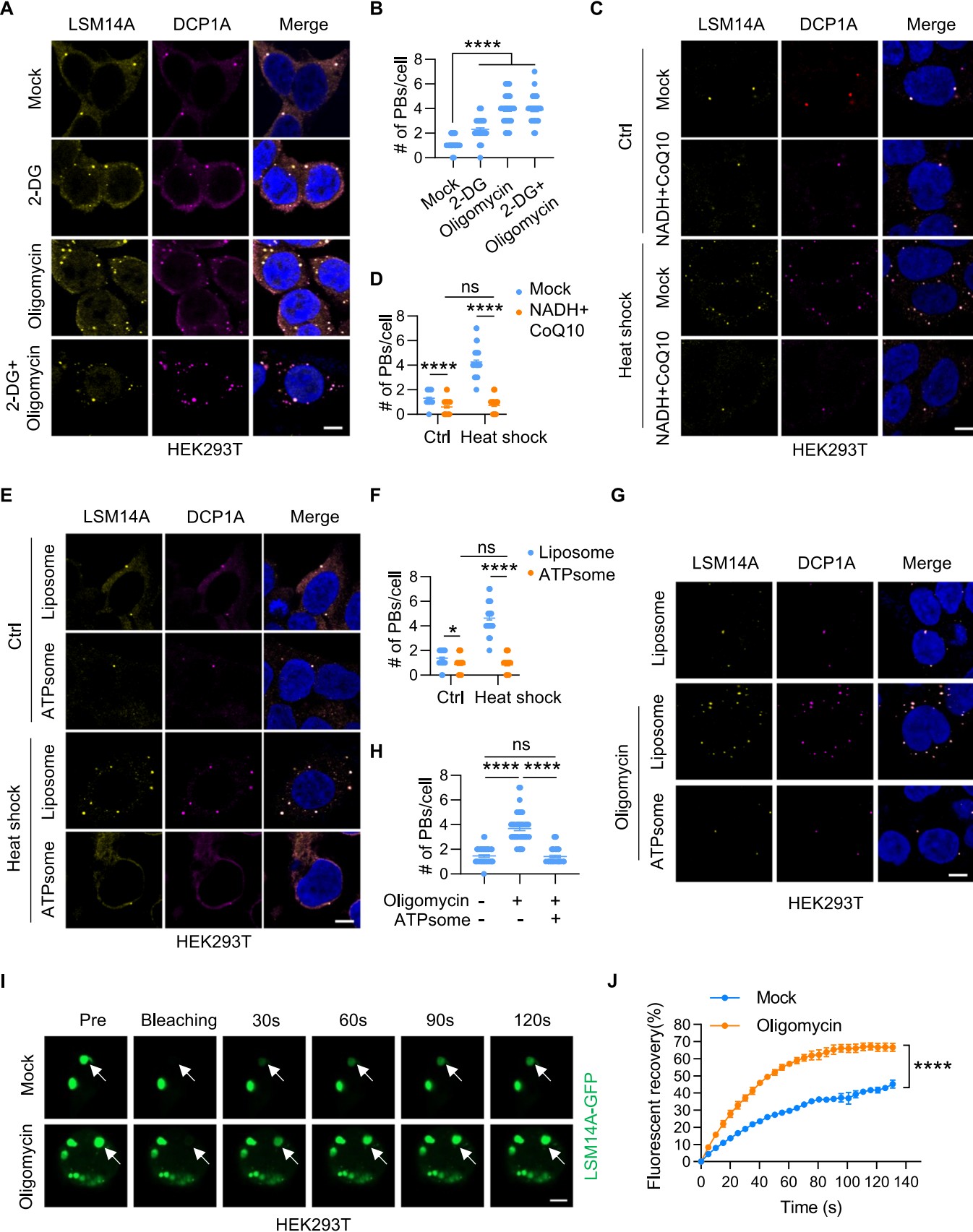

◄ **Figure 1. ATP depletion fosters P-bodies formation.**

(A, B) HEK293T cells were treated with 2-DG or/and oligomycin (2 h) and stained for LSM14A and DCP1A. Representative images are shown in (A). The number of P-bodies within each cell from one of three biological replicates is plotted in (B) ($n = 50$). Scale bar, 5 μm. Error bars indicate SEM. ****$P < 0.0001$ (Student's t test). (C, D) HEK293T cells treated with or without NADH+CoQ10 (2 h) were exposed to heat shock (42 °C) (1 h) and stained for LSM14A and DCP1A. Cells were imaged (C), and the number of P-bodies within each cell was quantified (D) ($n = 50$). Scale bar, 5 μm. Error bars indicate SEM. ns: no significance, ****$P < 0.0001$ (two-way ANOVA). (E, F) HEK293T cells treated with liposome or ATPsome (1 h) were exposed to heat shock (42 °C) and stained for LSM14A and DCP1A. Representative images are shown in (E). The number of P-bodies within each cell is plotted in (F) ($n = 50$). Scale bar, 5 μm. Error bars indicate SEM. *$P < 0.05$, ****$P < 0.0001$ (two-way ANOVA). (G, H) HEK293T cells subjected to different treatments were stained for LSM14A and DCP1A. Cells were imaged (G), and the number of P-bodies within each cell was quantified (H) ($n = 50$). Scale bar, 5 μm. Error bars indicate SEM. ns: no significance, ****$P < 0.0001$ (Student's t test). (I, J) HEK293T cells expressing LSM14A-GFP were treated with DMSO and oligomycin, and the relative mobility of LSM14A-GFP was determined by FRAP (I, J). White arrows indicate the target droplets for FRAP. Scale bar, 5 μm. $n = 3$ experiments, Error bars indicate SEM. ****$P < 0.0001$ (two-way ANOVA). Source data are available online for this figure.

## TRIM23 regulates P-bodies assembly induced by energy stress

Ubiquitin is attached to target proteins via a three-step process mediated by ubiquitin-activating enzymes (E1), ubiquitin-conjugation enzymes (E2), and ubiquitin ligases (E3) (Hershko and Ciechanover, 1992). With over 600 predicted E3 ligases in the human genome that confer specificity to the ubiquitin reaction (Kannt and Đikić, 2021), we sought to identify the E3 ubiquitin ligase responsible for energy stress-induced P-bodies assembly through a pooled CRISPR screen. LSM14A-GFP-expressing HEK293T cells were selected for screening, as the fluorescence intensity was positively correlated with the number of P-bodies (Fig. EV3A–C). Thus, we could isolate cells with more P-bodies from the cell population by fluorescence-activated cell sorting to determine the sgRNA distribution.

We then designed a sgRNA library targeting 620 E3 ubiquitin ligases using optimized sgRNA sequences from the CRISPick library, which contains a total of 2670 sgRNAs, including 200 negative control sgRNAs. Considering that the number of P-bodies in individual cell was heterogeneous, HEK293T cells with a moderate LSM14A-GFP fluorescence intensity, representing approximately two P-bodies per cell, were initially selected for screening. Following infection with the all-in-one lentiviral sgRNA library and puromycin selection, oligomycin was added to induce P-bodies formation, and HEK293T cells with high LSM14A-GFP fluorescence intensity were selected for sgRNA sequencing (Fig. 3A).

In our analysis, we identified 63 sgRNAs targeting 24 enriched genes and 71 sgRNAs targeting 27 depleted genes in LSM14A-GFP^high cells relative to unsorted input cells (Figs. 3B,C and Dataset EV1). Considering the positive role of ubiquitin on P-bodies formation, we hypothesized that the sgRNA targeting the regulator associated with the response to energy stress would be identified among the depleted sgRNAs. We tested ten top-ranking positive regulators using individual CRISPR knockouts and found that deletion of *LNX1*, *UBE3*, and *TRIM23* led to a reduction in the number of P-bodies (Fig. EV3D,E). Among these three regulators, we focused on TRIM23 (tripartite motif containing 23), a RING domain-containing E3 ligase (Vichi et al, 2005), given the recently documented role of another member of the TRIM family, TRIM24, in regulating P-bodies assembly (Wei et al, 2022). Knocking out TRIM23 decreased the number of P-bodies under both basal and ATP-depletion conditions in HEK293 cells and RKO cells (Figs. 3D,E and EV3F–I). Notably, oligomycin was unable to induce P-bodies formation in *TRIM23* knockout cells (Figs. 3D,E

and EV3F–I), demonstrating the dependence of energy stress-associated P-bodies on TRIM23. Similarly, the number of P-bodies remained intact in *TRIM23* knockout cells even when ubiquitin was overexpressed (Figs. 3F,G and EV3J,K). Conversely, TRIM23 overexpression promoted P-bodies formation in HEK293 cells and HCT116 cells (Figs. 3H,I and EV3L–O), mimicking the effects of oligomycin addition and forced expression of ubiquitin. Over-expressing TRIM23 had a limited effect on stress granule assembly under both basal and sodium arsenite conditions (Fig. EV3P,Q), suggesting that TRIM23 is a specific regulator of P-bodies. Furthermore, TRIM23 deficiency reduced the total recovery of LSM14A-GFP (Fig. 3J,K), while its overexpression promoted FRAP (Fig. 3L,M). These data illustrate that TRIM23 promotes P-bodies formation and dynamics, especially under energy stress.

## TRIM23 is a K63-specific ubiquitin ligase for HAX1

To investigate the mechanism underlying P-bodies formation mediated by TRIM23, we sought to identify the downstream target of TRIM23. We found that the TRIM23 protein was present as liquid droplets in the cytoplasm and colocalized with markers of the P-bodies (Fig. 4A), prompting us to focus on P-bodies proteins first. Interestingly, TRIM23 selectively interacted with HAX1 (HCLS1-associated protein X-1) (Fig. 4B), but not with other P-bodies proteins, such as LSM14A, DDX6, DCP1A, EDC3, and NBDY (Fig. EV4A). In accordance with previous data (Zayat et al, 2015), HAX1 colocalized with the P-bodies proteins (LSM14A, DDX6, 4E-T, and DCP1A) and TRIM23 (Figs. 4C and EV4B,C).

Treating HEK293T cells with oligomycin promoted the poly-ubiquitination of endogenous HAX1 and increased the protein level of TRIM23, without affecting the total protein level of HAX1 (Fig. 4D). Moreover, TRIM23 strongly induced polyubiquitination of both endogenous HAX1 and ectopically expressed Flag-HAX1, with no alteration in total protein levels (Figs. 4E,F and EV4D,E). Given the significance of K63 ubiquitin in P-bodies dynamics, we next determined whether TRIM23 links the K63 polyubiquitin chain to HAX1. As expected, K63R ubiquitin could not be linked to HAX1, even when TRIM23 was overexpressed, whereas wild-type and K48R ubiquitin still enhanced HAX1 polyubiquitination (Fig. 4G). Thus, TRIM23 is critical for K63 ubiquitination of HAX1.

TRIM23 contains four domains (RING, B-box, coiled-coil, and ARF), of which the RING domain has E3 ligase activity, and the ARF domain is responsible for interactions with other proteins (Sparrer et al, 2017). To test the role of different domains in HAX1 ubiquitination, we generated a series of TRIM23 deletion mutations

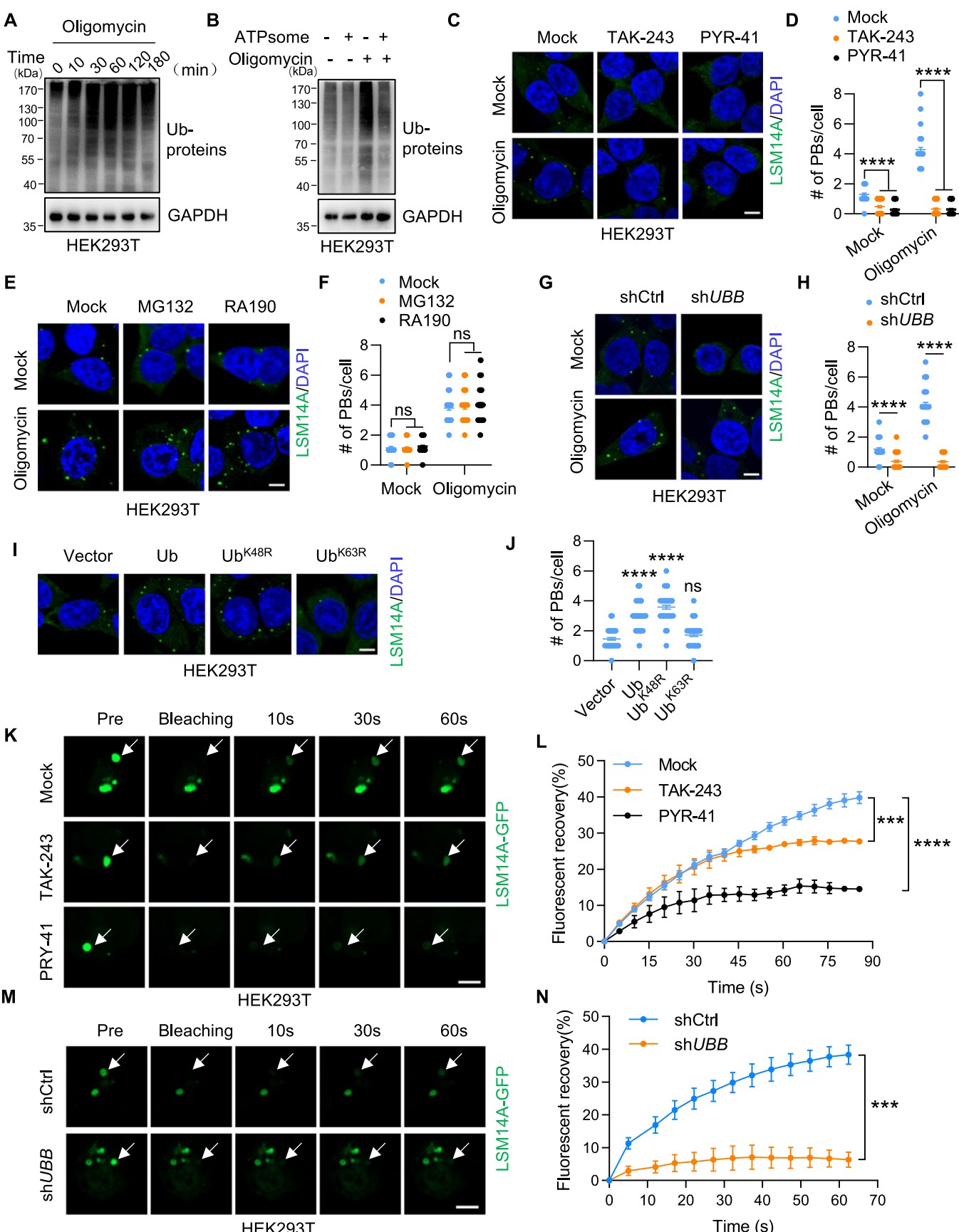

**Figure 2. ATP depletion-induced P-bodies depend on protein ubiquitination.**

(A, B) Immunoblot of HEK293T cells treated with oligomycin for the indicated times (**A**) or oligomycin with or without ATPsome (1 h) (**B**). (**C–F**) HEK293T cells were treated with TAK-243 or PYR-41 (1 h) (**C, D**), or MG132 or RA190 (1 h) (**E, F**) in the presence or absence of oligomycin (1 h) and stained for LSM14A and DAPI. Representative images are shown in (**C, E**). The number of P-bodies within each cell is plotted in (**D, F**) ($n = 50$). Scale bar, 5 µm. Error bars indicate SEM. ns: no significance, ****$P < 0.0001$ (two-way ANOVA). (**G, H**) Control and *UBB*-knockdown HEK293T cells were treated with or without oligomycin (1 h) and stained for LSM14A and DAPI. Representative images are shown in (**G**). The number of P-bodies within each cell is plotted in (**H**) ($n = 50$). Scale bar, 5 µm. Error bars indicate SEM. ****$P < 0.0001$ (two-way ANOVA). (**I, J**) HEK293T cells were transfected with vector, Ub, Ub$^{K48R}$ or Ub$^{K63R}$ and stained for LSM14A and DAPI. Representative images are shown in (**I**). The number of P-bodies within each cell is plotted in (**J**) ($n = 50$). Scale bar, 5 µm. Error bars indicate SEM. ****$P < 0.0001$ (Student's t test). (**K–N**) HEK293T cells expressing LSM14A-GFP were treated with DMSO, TAK-243 or PYR-41 (1 h) (**K, L**) or transfected with shCtrl or sh*UBB* (**M, N**). The relative mobility of LSM14A-GFP was determined by FRAP. White arrows indicate the target droplets for FRAP. Scale bar, 5 µm. $n = 3$ experiments, Error bars indicate SEM. ***$P < 0.001$, ****$P < 0.0001$ (two-way ANOVA). Source data are available online for this figure.

(Fig. 4H). TRIM23ΔA and TRIM23ΔRA could not interact with HAX1, while TRIM23ΔRBC retained this ability, confirming that the ARF domain is critical for the binding of TRIM23 to HAX1 (Fig. 4I). Furthermore, TRIM23 lacking either the ARF domain (ΔA, ΔRA) or the RING domain (ΔR, ΔRB, ΔRBC) could not catalyze HAX1 ubiquitination, while deletion of the B-box and coiled-coil domains had little effect on the polyubiquitination of HAX1 (Fig. 4I). Consistently, loss of the ARF domain (ΔA, ΔRA) or RING domain (ΔR, ΔRB, ΔRBC) deprived the function of TRIM23 in P-bodies formation (Figs. 4J and EV4F).

Five ub-lysines in HAX1 were identified by mass spectrometry (MS) according to the databases of post-translational modifications (Hornbeck et al, 2015) (Fig. EV4G). To determine the ubiquitination site(s) on HAX1, we mutated all five candidate Lys residues to Arg individually. Among these mutations, only the K131R mutant exhibited almost no ubiquitination when TRIM23 was overexpressed, but the K131R mutation of HAX1 did not affect its interaction with TRIM23 (Fig. 4K). These data indicate that TRIM23 interacts with HAX1 via the ARF domain, attaching the K63 polyubiquitin chain to the K131 residue of HAX1.

## Ubiquitination of HAX1 is required for P-bodies assembly

To examine the importance of HAX1 in P-bodies formation, we knocked down HAX1 in HEK293T cells, and found a decreased number of LSM14A-labeled P-bodies (Fig. EV5A–C). Furthermore, we used CRISPR/Cas9 to knock out *HAX1* in HEK293T cells (hereafter referred to as *HAX1* KO cells) (Fig. EV5D). To our surprise, no P-bodies could be detected in *HAX1* KO cells under both basal and ATP-depletion conditions (Fig. 5A,B), despite the consistent expression of the conserved core proteins within P-bodies (Fig. EV5D). Similar to TRIM23, HAX1 knockout did not affect stress granule formation under either basal or sodium arsenite conditions (Fig. EV5E,F). Moreover, expression of wild-type HAX1, but not the ubiquitination-deficient mutant HAX1 (K131R), was sufficient to restore P-bodies formation in *HAX1* KO cells under both basal and ATP-depletion conditions (Fig. 5C,D), indicating the essential role of HAX1 ubiquitination in supporting P-bodies assembly.

## Ubiquitinated HAX1 is essential for the condensation of LSM14A and DDX6

To explore how ubiquitinated HAX1 regulates P-bodies assembly induced by energy stress, we monitored the condensation process

of HAX1 and P-bodies core proteins (DCP1A, LSM14A, and DDX6) during P-bodies formation. To this end, we fractionated cell extracts into soluble (S2) and particular (P2) fractions, as described previously (Rzeczkowski et al, 2011; Teixeira et al, 2005), to distinguish soluble cytoplasm from insoluble condensates containing partially purified P-bodies. First, we stimulated HEK293T cells with oligomycin for different times and performed cell fractionation to dynamically determine the protein distribution. The distribution of HAX1 and DCP1A changed rapidly, with an immediate reduction in the cytosolic fraction and an augmentation in the P2 fraction after only five minutes oligomycin treatment (Figs. 5E and EV5G). The visible changes in DDX6 and LSM14A were detectable at fifteen and thirty minutes, respectively, much later than changes in HAX1 and DCP1A (Figs. 5E and EV5G), suggesting that HAX1 is a pioneer protein that condenses during P-bodies formation.

Similarly, TRIM23 elevated the P2 fractions of HAX1 and P-bodies core proteins (DCP1A, LSM14A, and DDX6) (Figs. 5F and EV5H), suggesting that TRIM23 facilitates these core proteins concentration. Neither LSM14A nor DDX6 could shift from the cytoplasm to the P-bodies enriched pellet induced by TRIM23 or oligomycin in both *HAX1* KO (Fig. 5F,G) and knockdown cells (Fig. EV5I,J). Moreover, HAX1$^{K131R}$ not only lost its ability to translocate into P2 fractions but also suppressed the condensation of LSM14A and DDX6 (Figs. 5H,I and EV5H). Knockdown of HAX1 did not reduce the DCP1A condensation induced by TRIM23 or oligomycin (Fig. EV5I,J), and there was still a small amount of DCP1A in the P2 fraction in *HAX1* KO and HAX1$^{K131R}$ cells (Fig. 5F–I), illustrating that the change in DCP1A during P-bodies formation is independent of HAX1. These results suggested that the condensation of LSM14A and DDX6, which is the key process of P-bodies formation, is dependent on the ubiquitination of HAX1.

Consistent with these findings, endogenous HAX1 coimmunoprecipitated with LSM14A and DDX6 but not with DCP1A, and these interactions were significantly strengthened upon oligomycin stimulation (Fig. 5J). TAK-243 treatment decreased the interaction of HAX1 with LSM14A and DDX6 (Fig. 5J), suggesting that the interaction is ubiquitin-dependent. Similarly, HAX1$^{K131R}$ showed an evidently reduced interaction with LSM14A and DDX6, which could not be reversed by oligomycin stimulation (Fig. 5K). These results illustrate that the condensation of HAX1 is an early event in P-bodies assembly induced by energy stress and that this ubiquitination-dependent process is indispensable for the condensation of LSM14A and DDX6.

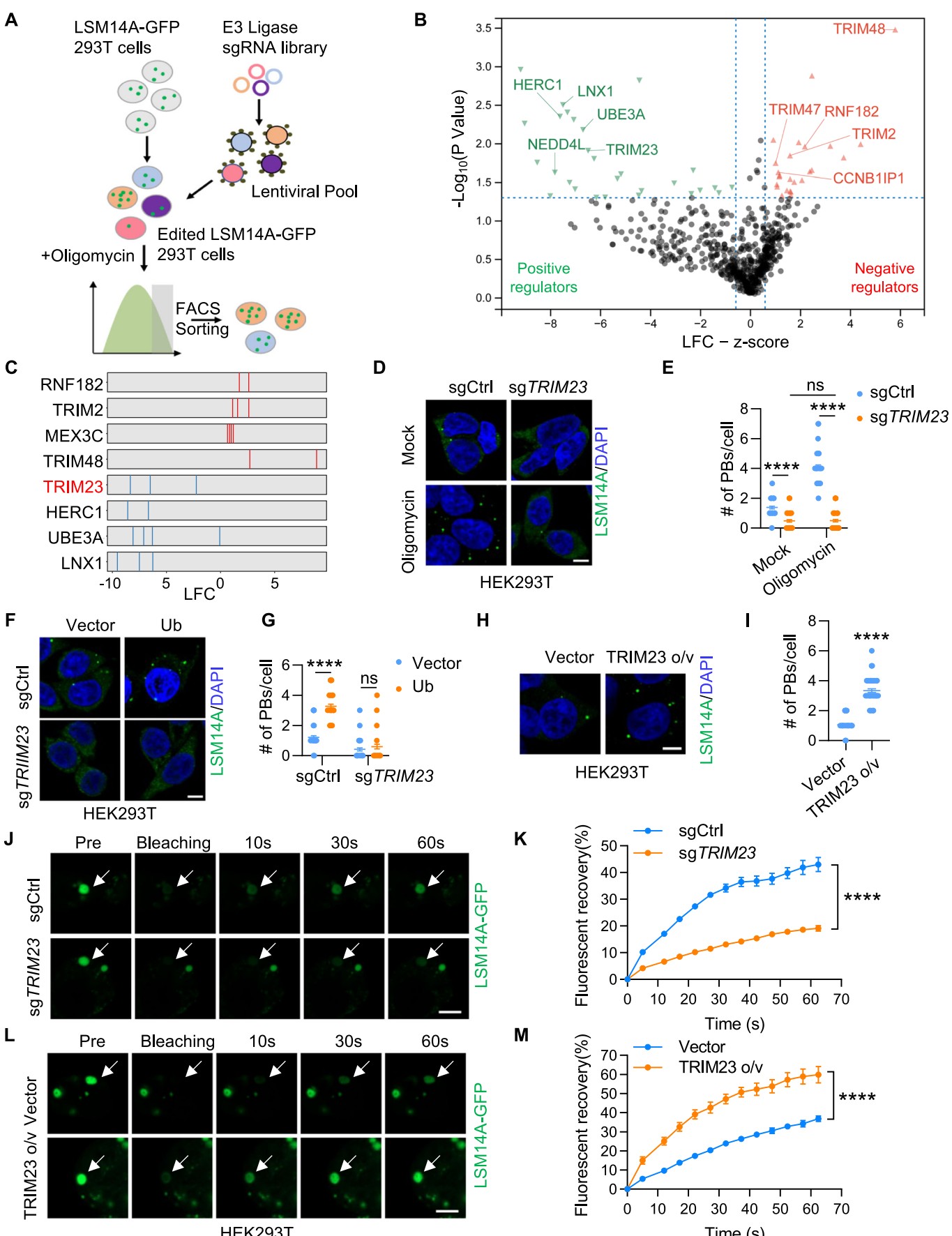

◄

**Figure 3.   Energy stress regulates P-bodies assembly via TRIM23.**

(A) Schematic of pooled CRISPR E3 ligase screening. (B) Volcano plot for hits from the E3 ligase screening. The x-axis shows the z-score for the gene-level log2-fold change (LFC); the median LFC for all sgRNAs per gene was scaled. The y-axis shows the $P$ value calculated by MAGeCK. The red dots indicate negative regulators (enriched in LSM14A-GFP^high cells), whereas the blue dots indicate positive regulators (depleted in LSM14A-GFP^high cells) defined by a $P$ value < 0.05 (Permutation test) and an LFC > 2. (C) Distribution of individual sgRNAs for representative genes enriched (red lines) and depleted (blue lines) in LSM14A-GFP^high cells. (D, E) *TRIM23* knockout and control HEK293T cells were treated with oligomycin (1 h) and immunolabeled for LSM14A and DAPI. Cells were imaged (D), and the number of P-bodies within each cell was quantified (E) ($n = 50$). Scale bar, 5 μm. Error bars indicate SEM. ns: no significance, ****$P < 0.0001$ (two-way ANOVA). (F, G) *TRIM23* knockout and control HEK293T cells were transfected with vector or Ub and stained for LSM14A and DAPI. Cells were imaged (F), and the number of P-bodies within each cell was quantified (G) ($n = 50$). Scale bar, 5 μm. Error bars indicate SEM. ns: no significance, ****$P < 0.0001$ (two-way ANOVA). (H, I) HEK293T cells were transfected with vector or TRIM23 and stained for LSM14A and DAPI. Cells were imaged (H), and the number of P-bodies within each cell was quantified (I) ($n = 50$). Scale bar, 5 μm. Error bars indicate SEM. ****$P < 0.0001$ (Student's t test). (J–M) FRAP analysis of GFP-LSM14A in HEK293T cells with knockout of *TRIM23* (J, K) or overexpression of TRIM23 (L, M). White arrows indicate the target droplets for FRAP. Scale bar, 5 μm. $n = 3$ experiments, Error bars indicate SEM. ****$P < 0.0001$ (two-way ANOVA). Source data are available online for this figure.

## HAX1 undergoes ubiquitination-dependent phase separation

LLPS is the important mechanism driving condensation of proteins into membraneless organelles. Many P-bodies proteins possessing LLPS ability harbor IDRs, suggesting their significance in the regulation of P-bodies assembly (Lin et al, 2017; Wheeler et al, 2016). Given that HAX1 also has multiple IDRs (Fig. 6A), we next sought to examine whether the HAX1 protein alone undergoes LLPS. We observed that at a physiological salt concentration (150 mM NaCl), purified HAX1 proteins displayed concentration-dependent LLPS (Fig. 6B; Appendix Fig. S1A). Direct supplement of ATP had negligible impact on HAX1 droplets formation (Appendix Fig. S1B), suggesting that the hydrotropic effects of ATP is not involved in HAX1 LLPS. Moreover, droplets of HAX1 exhibited fusion behavior in HEK293T cells (Fig. 6C), and recovered most fluorescence signals within ~120 s after photobleaching (Fig. 6D,E), confirming the occurrence of HAX1 LLPS in cells.

To gain insights into which IDRs of HAX1 support LLPS, we generated constructs lacking one IDR in every truncated version and expressed them in *HAX1* KO cells. Interestingly, only IDR2, which contains the ubiquitination site of HAX1, was indispensable for HAX1 puncta formation and P-bodies assembly (Fig. 6F,G). Furthermore, HAX1^K131R lost the capability for droplet formation and broke LSM14A-labeled P-bodies under basal conditions (Fig. 6H,I). Additionally, oligomycin treatment (Fig. 6H,I) and TRIM23 overexpression (Fig. 6J,K) did not induce puncta formation of HAX1^K131R, proving that the energy stress-induced LLPS of HAX1 completely depended on ubiquitination.

To establish the relationship between LLPS of HAX1 and P-bodies formation, we added the FUS N-terminal IDR region (IDR^FUS) to the HAX1^K131R and HAX1^ΔIDR2 mutants to forcedly induce LLPS. The HAX1^ΔIDR2-IDR^FUS and HAX1^K131R-IDR^FUS fusion proteins formed round condensates in the cytoplasm, and P-bodies labeled with LSM14A also reformed (Appendix Fig. S1C–F), suggesting the dependence of P-bodies formation on LLPS of HAX1.

Next, we investigated the roles of ubiquitin and TRIM23 in HAX1 LLPS. Consistent with the finding that K63 polyubiquitination of HAX1 regulates P-bodies assembly, ubiquitin knockdown suppressed mCherry-HAX1 FRAP (Fig. 6L; Appendix Fig. S1G), while ubiquitin overexpression facilitated mCherry-HAX1 FRAP (Fig. 6M; Appendix Fig. S1H). Additionally, we confirmed that

TRIM23 affected the LLPS ability of HAX1 (Fig. 6N,O; Appendix Fig. S1I,J). These findings support our hypothesis that TRIM23-mediated polyubiquitination regulates HAX1 LLPS.

## TRIM23/HAX1 restrains protein synthesis

Recent studies have shown that P-bodies are important for translational repression (Horvathova et al, 2017; Hubstenberger et al, 2017). Accordingly, we explored whether TRIM23/HAX1 regulate protein translation in mammalian cells in response to energy stress. The surface sensing of translation (SUnSET) method, which relies on the incorporation of puromycin into elongating peptide chains and can then be detected immunologically using an anti-puromycin antibody, was used as an indicator of bulk translation activity (Schmidt et al, 2009). We found that TRIM23 suppressed overall protein synthesis in both HEK293T and CRC cells (Fig. 7A–D). Similarly, the knockout of HAX1 promoted overall protein synthesis (Fig. 7E), while overexpressing WT HAX1, but not the K131R mutant HAX1, resulted in the suppression of protein synthesis (Fig. 7F). Oligomycin treatment suppressed protein synthesis in WT cells, but this effect was absent in *TRIM23* KO (Fig. 7C,D) or *HAX1* KO cells (Fig. 7E). Moreover, the addition of FUS^IDR (Murray et al, 2017) restored the ability of HAX1^ΔIDR2 and HAX1^K131R to inhibit protein synthesis (Fig. 7G,H). These results suggest that TRIM23/HAX1 is essential for translational inhibition induced by energy stress, which is mediated by P-bodies.

## A tumorigenic role of ubiquitinated HAX1 in colorectal cancer

Mounting evidence suggests that cancer cells metabolize glucose by anaerobic glycolysis, a less efficient way of ATP generation compared to oxidative phosphorylation (Vander Heiden et al, 2009). Rapid cell proliferation and high rates of protein synthesis in cancer cells result in excessive ATP consumption and consequently lower intracellular ATP levels. We observed that ATP concentrations were lower in CRC cells than in normal colon epithelial cells (Appendix Fig. S2A). Furthermore, it has been proven that P-bodies components are closely related to the occurrence and progression of various tumors (Hardy et al, 2017; Wu et al, 2018). We confirmed that P-bodies were significantly increased in CRC cell lines and patient tissues (Appendix Fig. S2B,C), prompting us to investigate the role of the TRIM23/HAX1 pathway in CRC.

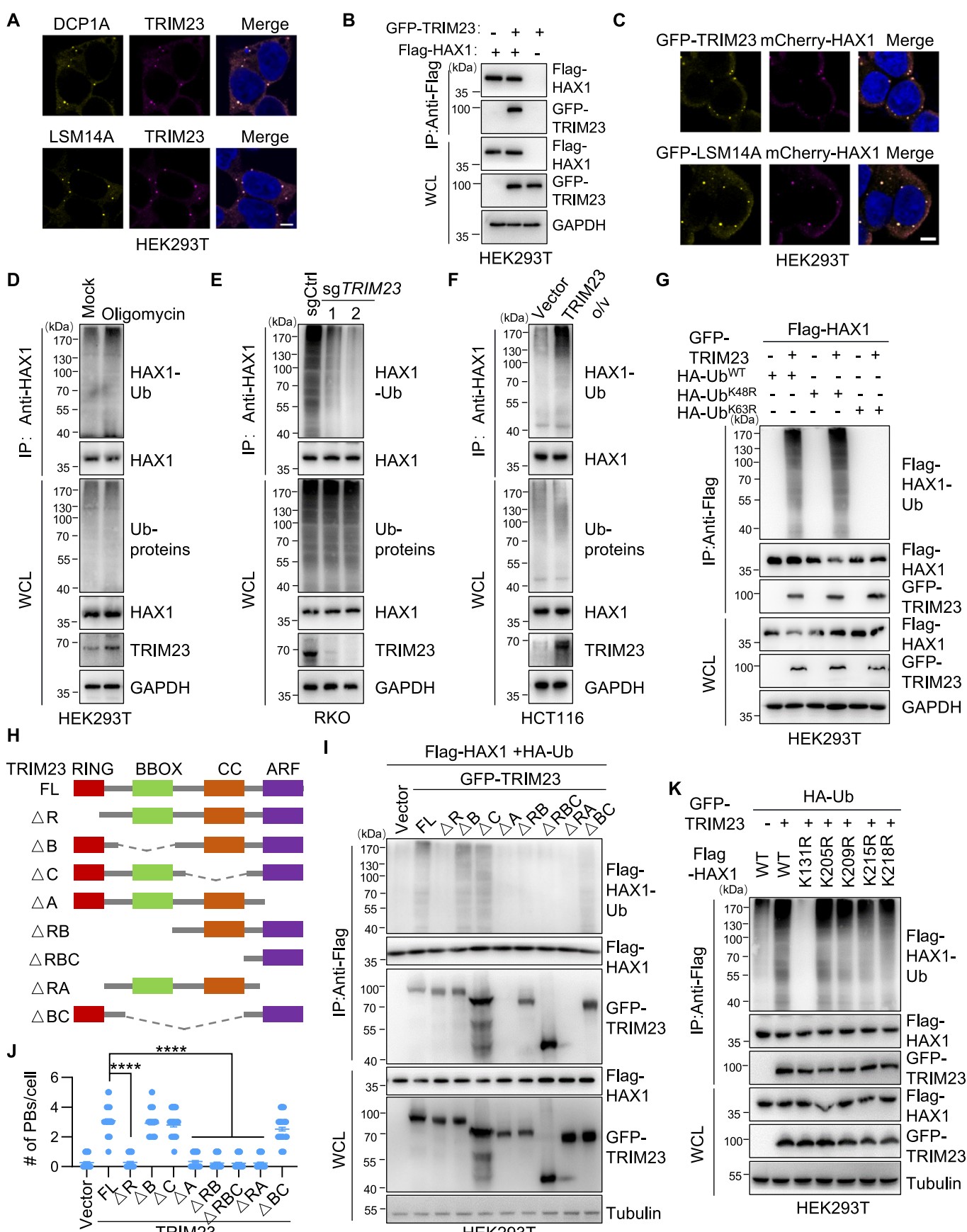

**Figure 4. TRIM23 ligases K63 ubiquitin to specific Lys residue of HAX1.**

(A) HEK293T cells were fluorescently imaged for TRIM23 and DCP1A (above) or LSM14A (below). Scale bars: 5 μm. (B) HEK293T cells were transfected with Flag–HAX1 and GFP–TRIM23 as indicated. The HAX1–TRIM23 interaction was examined by co-IP. (C) Fluorescence imaging of HEK293T cells transfected with mCherry-HAX1 and GFP-TRIM23 (above) or GFP-LSM14A (below). Scale bars: 5 μm. (D) Immunoblot of HEK293T cells extracts captured with an antibody against HAX1 after exposure to oligomycin or DMSO for 1 h. (E) Immunoblot of control and *TRIM23*-knockout RKO cells extracts captured with antibody to HAX1. (F) Immunoblot of control and TRIM23-overexpressing HCT116 cells extracts captured with antibody to HAX1. (G) Immunoblot of transfected HEK293T cells extracts captured with antibody to Flag-tag. HEK293T cells were transfected with Flag-HAX1 and GFP-TRIM23 together with wild-type ubiquitin (Ub$^{WT}$) or mutant ubiquitin (Ub$^{K48R}$ or Ub$^{K63R}$). (H) Schematic of full-length TRIM23 and domain deletion mutants. (I) Lysates from HEK293T cells transfected with Flag-HAX1, HA-Ub, and TRIM23 (FL or its deletion mutants) were analyzed via immunoprecipitation with an anti-Flag antibody, followed by immunoblotting. FL, full length; ΔR, deletion of R domain. (J) HEK293T cells were transfected with vector-GFP, TRIM23-GFP-FL or its deletion mutants and stained for DCP1A and DAPI. Representative images are shown in Fig. EV4F. The number of P-bodies represented by DCP1A within each cell is plotted in (J) (*n* = 50). Error bars indicate SEM. ****$P < 0.0001$ (Student's t test). (K) Immunoblot of transfected HEK293T cells extracts captured with antibody to Flag-tag. HEK293T cells were transfected with HA-Ub, GFP-TRIM23, and WT Flag-HAX1 or its mutants. Source data are available online for this figure.

Through immunohistochemistry (IHC) analysis of clinical samples of human CRC tissues and paired normal intestinal tissues, we found that the protein level of TRIM23 was slightly lower in CRC tissues (Fig. 8A; Appendix Fig. S2D), while there was no correlation between the protein level of TRIM23 and overall survival of patients (Appendix Fig. S2E). Interestingly, we noticed that the TRIM23 protein was located in the nucleus of most normal tissues (101/106) and translocated into the cytoplasm in ~30% of CRC tissues (30/106) (Fig. 8A,B), consistent with the distribution pattern observed in CRC cell lines and normal controls (Appendix Fig. S2F). Moreover, high cytoplasmic TRIM23 expression was correlated with poor survival in patients with CRC (Fig. 8C). Similarly, HAX1 protein was slightly induced in CRC tissues (Appendix Fig. S2G), and a high protein level of HAX1 was correlated with poor survival of patients (Appendix Fig. S2H). Due to the lack of a specific antibody for ubiquitinated HAX1, we defined ubiquitination levels based on TRIM23 localization and HAX1 protein levels, categorizing CRC tissues into four clinical-pathological types (Fig. 8D). We considered high levels of HAX1 along with the cytoplasmic location of TRIM23 as the surrogate marker for high HAX1 ubiquitination (HAX1$^H$TRIM23$^C$) and low levels of HAX1 along with nuclear location of TRIM23 as low HAX1 ubiquitination (HAX1$^L$TRIM23$^N$), which were correlated with the best and worst survival outcomes, respectively (Fig. 8E).

Moreover, knocking out TRIM23 suppressed RKO cells proliferation, while overexpressing cytoplasmic TRIM23, but not nuclear TRIM23 (TRIM23-NLS), promoted the proliferation of HCT116 cells (Fig. 8F,G; Appendix Fig. S2I,J). HAX1-knockout HCT116 cells exhibited weakened proliferation, which was rescued by WT HAX1 but not by ubiquitination-deficient HAX1 (K131R) (Fig. 8H,I; Appendix Fig. S2K,L). In accordance with these in vitro results, xenograft model results demonstrated that the progression of tumors and P-bodies formation were also controlled by cytoplasmic TRIM23 and ubiquitinated HAX1 (Fig. 8J–M; Appendix Fig. S2M–P). Collectively, these results showed that the TRIM23/HAX pathway was critical for the tumorigenicity of CRC.

Finally, we investigated the dependence of TRIM23/HAX1-mediated CRC tumorigenicity on P-bodies. Overexpression of TRIM23 in LSM14A-knockout HCT116 cells did not induce P-bodies formation (Appendix Fig. S3A,B) and did not promote cell proliferation in vitro (Fig. 8N). Reintroduction of HAX1 into the LSM14A/HAX1 double-knockout HCT116 cells also did not support the formation of P-bodies labeled with DDX6 and DCP1A, although the HAX1 granules were normal (Appendix Fig. S3C–E).

Reintroduction of HAX1 did not promote the proliferation of LSM14A/HAX1 double-knockout HCT116 cells (Fig. 8O). In addition, disruption of P-bodies with 1,6-hexanediol (1,6-HD), an aliphatic alcohol disassembling LLPS-dependent macromolecular condensate (Liu et al, 2021; Strom et al, 2017), inhibited the ability of TRIM23 and HAX1 to promote HCT116 cell proliferation (Appendix Fig. S3F–K). Taken together, these data provide evidence that the function of TRIM23/HAX1 in the tumorigenicity of CRC depends on P-bodies.

## Discussion

The cellular energy state, characterized by the intracellular level of adenosine triphosphate (ATP), plays a pivotal role in the dynamics of RNP granules. However, the specific role of ATP seems to be context-specific. For instance, while ATP deficiency can impair sodium arsenite-induced stress granule formation (Jain et al, 2016), it also promotes glycolytic inhibition-induced stress granule assembly (Wang et al, 2022). In addition, ATP is required for the disassembly of both P-bodies and stress granules, supporting its role as a biological hydrotrope in maintaining protein solubility and preventing macromolecular aggregation through LLPS (Patel et al, 2017). However, the precise role and underlying mechanism of ATP in P-bodies remain undefined. Our data demonstrated that ATP depletion-induced P-bodies formation through the TRIM23-mediated ubiquitin pathway. TRIM23 acts as a specific E3 ligase for HAX1, which is also a specific regulator of the P-bodies induced by ATP depletion.

As important RNPs condense in the cytoplasm, P-bodies and stress granules share many similar properties. For example, both foci can be induced by stress stimulation (e.g., arsenide, heat shock) and contain overlapping protein components (Youn et al, 2019). In this study, we found that ATP had disparate functions in these two biomolecular condensates in vivo. This finding suggested that cells respond to different external stress stimuli by producing different RNP condensates and that P-bodies, but not stress granules, are characteristic membraneless organelles for cells to cope with ATP deficiency.

While mass spectrometry (MS)-based proteomics have been applied for the global identification of P-bodies proteins (Hubstenberger et al, 2017; Xing et al, 2020; Youn et al, 2019), the complete composition remains elusive due to cell type and condition-dependent variability. Although HAX1 has not been

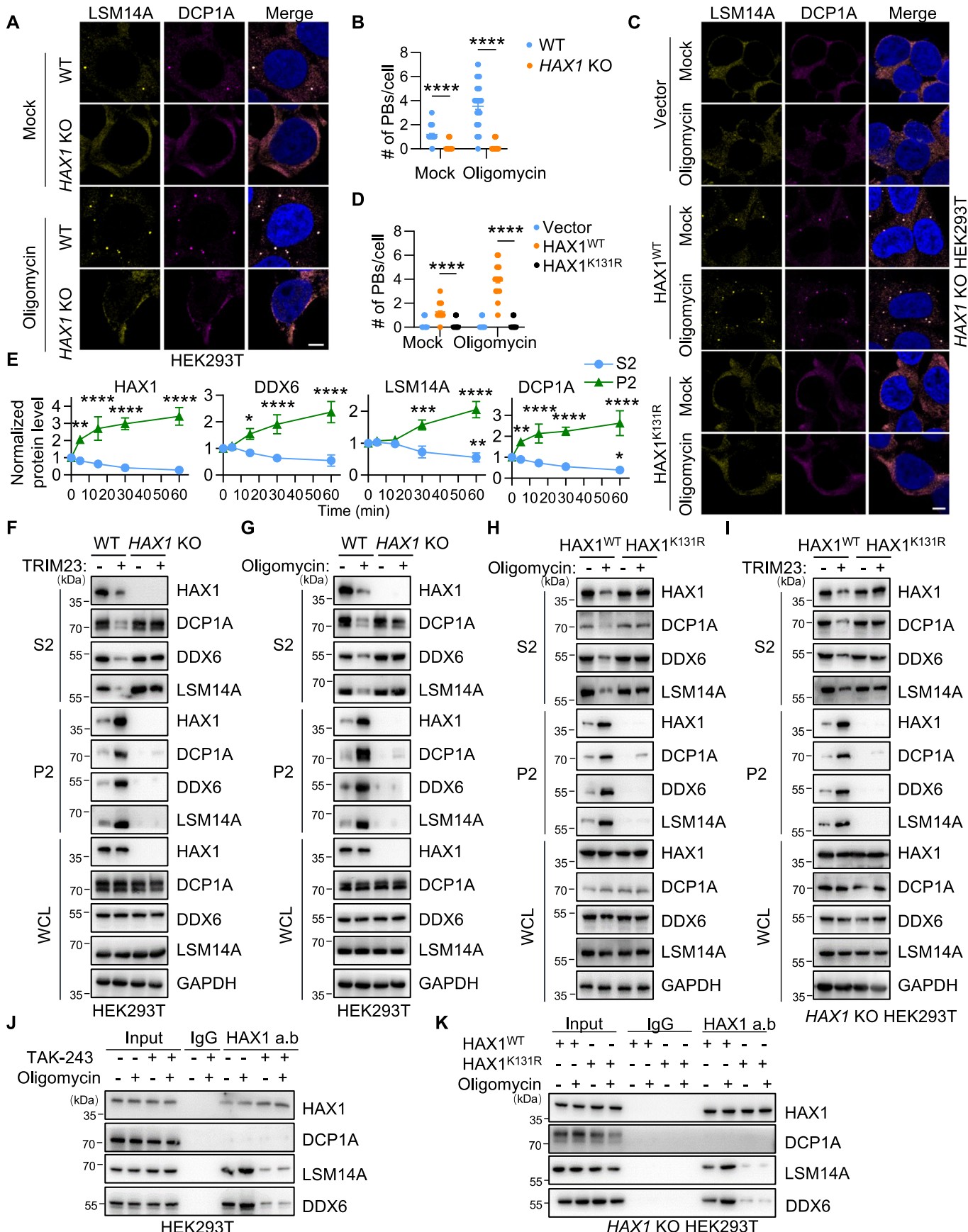

**Figure 5.  Ubiquitinated HAX1 triggers P-bodies assembly.**

(A, B) WT and *HAX1* KO HEK293T cells were treated with oligomycin or DMSO (1 h) and stained for the P-body markers LSM14A and DCP1A. Representative images are shown in (A). Scale bar, 5 μm. The number of P-bodies represented by DCP1A/LSM14A within each cell is plotted in (B) ($n = 50$). Error bars indicate SEM. ****$P < 0.0001$ (two-way ANOVA). (C, D) *HAX1* KO HEK293T cells were transfected with HAX1$^{WT}$ or HAX1$^{K131R}$, treated with oligomycin or DMSO (1 h) and stained for the P-body markers LSM14A and DCP1A. Representative images are shown in (C). Scale bar: 5 μm. The number of P-bodies within each cell is plotted in (D) ($n = 50$). Error bars indicate SEM. ****$P < 0.0001$ (two-way ANOVA). (E) S2 and P2 fractions from HEK293T cells treated with oligomycin for the indicated times were analyzed by immunoblotting. Representative immunoblots are shown in Fig. EV5G. The relative quantification of immunoblots for HAX1, DDX6, LSM14A, and DCP1A from three biological replicates is shown. Error bars indicate SD. *$P < 0.05$, **$P < 0.01$, ***$P < 0.001$, ****$P < 0.0001$ (two-way ANOVA). (F, G) S2 and P2 fractions from WT or *HAX1* KO HEK293T cells with or without TRIM23 overexpression (F) or oligomycin treatment (1 h) (G) were analyzed by immunoblotting. (H, I) S2 and P2 fractions from *HAX1* KO HEK293T cells transfected with HAX1$^{WT}$ or HAX1$^{K131R}$ with or without oligomycin treatment (1 h) (H) or TRIM23 overexpression (I) were analyzed by immunoblotting. (J) HEK293T cells were treated with DMSO or oligomycin in the presence or absence of TAK-243 (1 h). Cell lysates were captured with magnetic beads conjugated with IgG or anti-HAX1 antibodies for IP, followed by immunoblotting. (K) *HAX1* KO HEK293T cells transfected with HAX1$^{WT}$ or HAX1$^{K131R}$ were treated with or without oligomycin for 1 h. Cell lysates were immunoprecipitated with control IgG or an anti-HAX1 antibody prior to immunoblotting. Source data are available online for this figure.

identified as a P-bodies protein in proteomics studies, it was confirmed to be located within P-bodies with a potential role in mRNA processing but has limited influence on P-bodies formation (Grzybowska et al, 2013; Zayat et al, 2015). Our results revealed that (ubiquitinated) HAX1 can affect P-bodies dynamics exclusively under conditions of ATP depletion, suggesting that the function of P-bodies proteins is also context-dependent.

P-bodies store and degrade mRNA, depending on multiple core proteins (e.g., DCP1/DCP2, DDX6, EDC4, LSM1-7, etc.), but these proteins may not assemble simultaneously. Previous studies have focused predominantly on identifying new P-bodies components or studying interactions between components, neglecting the assembly process of P-bodies. Our study reveals the assembly dynamics of P-bodies in the early stage of energy stress. We propose a model, like G3BP1 serving as the central node in stress granules (Yang et al, 2020) HAX1 might be the key scaffold molecule of P-bodies, which is the initiating and central component in the "scaffold/client" model.

Cancer cells metabolize glucose by aerobic glycolysis, an inefficient way to generate ATP, and consume high amounts of energy to sustain rapid cell proliferation and growth, resulting in low intracellular ATP levels. The hypoxic state in solid tumors further exacerbates energy stress and is closely associated with poor prognosis in various cancer patients (Singleton et al, 2021), while elevated ATP has been shown to impair tumor growth (Naguib et al, 2018). These observations suggest that ATP depletion might be one of the drivers of tumorigenesis, not just the outcome. Despite this, the precise role and physiological significance of ATP depletion in cancers remain elusive. Our results suggested that low ATP levels enhance the tumorigenicity of CRC through the TRIM23/HAX1/P-bodies pathway, providing new insights into the pathogenesis of CRC.

## Methods

### Plasmids

pSin-Flag-TRIM23, pSin-Flag-HAX1, pSin-Flag-DCP1A, pSin-Flag-EDC3 and pSin-Flag-NBDY were generated by amplifying human gene into the pSIN/EF1α-IRES-Puro lentiviral expression vector (pSin). pSin-GFP-LSM14A, pSin-mCherry-HAX1, and pSin-GFP-TRIM23 were generated by overlapping PCR. PIG-TRIM23

and PIG-TRIM23-NLS, used to overexpress TRIM23 and TRIM23-NLS in CRC cells via retroviral transduction, were constructed by subcloning the TRIM23 and TRIM23-NLS sequence into the MSCV-PIG vector. HA-Ub amplified from human cDNA, and HA-Ub$^{K48R}$ and HA-Ub$^{K63R}$ amplified from pet14b-Ubiquitin K48R/K63R (BRICS, SP-637/638) were inserted into pSin vector. The pSin-GFP-TRIM23 deletion mutants ΔR, ΔB, ΔC, ΔA, ΔRB, ΔRBC, ΔRA and ΔBC lacked the amino acids 82–231, 367–669, 676–1110, 1216–1722, 82–669, 82–1110, 82–231 + 1216–1722, and 367–1110, and pSin-mCherry-HAX1 deletion mutants ΔIDR1, ΔIDR2, ΔIDR3 and ΔIDR4 lacked the amino acids 45–195, 294–408, 408–645, and 651–786, were generated by overlapping PCR. The pSin-Flag-HAX1 mutants (K131R, K205R, K209R, K215R, and K218R) and pSin-mCherry-HAX1$^{K131R}$ were generated by site-directed mutagenesis. HAX1$^{ΔIDR2}$-IDR$^{FUS}$ and HAX1$^{K131R}$-IDR$^{FUS}$ were generated by cloning human IDR$^{FUS}$ sequence into the pSin-mCherry-HAX1$^{ΔIDR2}$ and pSin-mCherry-HAX1$^{K131R}$. LentiCRISPR v2 (Addgene plasmid, #52961) plasmids were used for E3 ligase screening and knockout assays. pLKO.1 shRNA plasmids were obtained from Sigma Aldrich, and used for knockdown assays. The sequence of sgRNAs and shRNAs were listed in Table EV1. All the constructs generated in this study were verified via DNA sequencing.

### Antibodies

Antibodies against the following proteins were used: TRIM23 (Proteintech, 12607-1-AP, for immunofluorescence assay), TRIM23 (ABclonal, A8329, for Western blot and IP), TRIM23 (Sigma, A91882, for Immunohistochemical staining), HAX1 (Proteintech, 11266-1-AP), LSM14A (Proteintech, 18336-1-AP), DDX6 (Proteintech, 14632-1-AP), HA (Proteintech, 51064-AP), LaminB1 (Proteintech, 12987-1-AP), G3BP2 (Proteintech, 16276-1-AP), TIA1 (Proteintech, 12133-2-AP), DCP1A (ABclonal, A7376), 4E-T (ABclonal, A15175), and (ABclonal, AE012), ubiquitin (CST, 43124 S), flag (CST, 14793S), GAPDH (Zsbio, TA-08) and β-Tubulin (Zsbio, TA-10).

### Cell culture

HEK293T cells were cultured in DMEM (HyClone, SH30243.01). RKO, DLD1, and HCT116 cells were cultured in RPMI 1640 (HyClone, SH30027.01). The culture media were supplemented

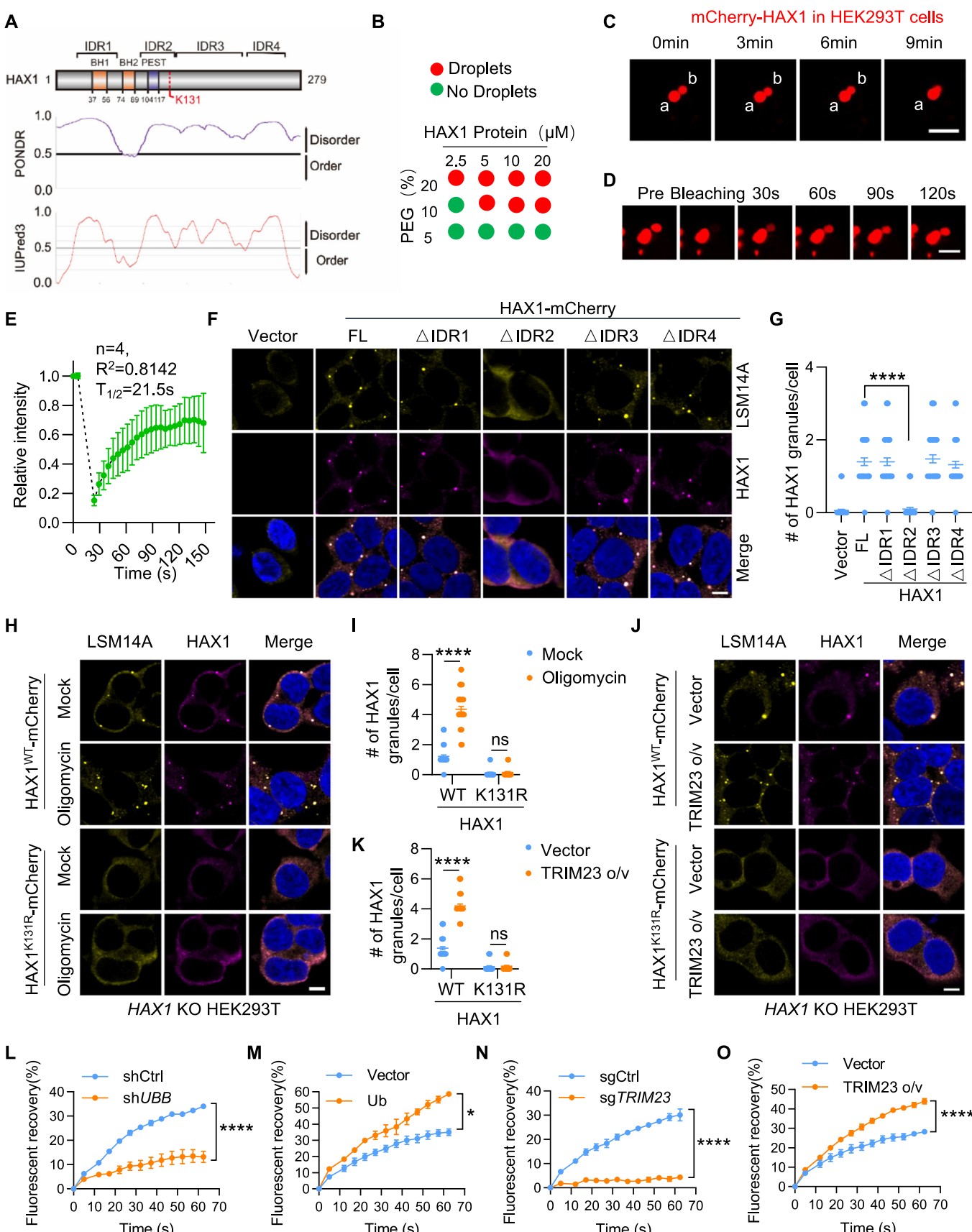

**Figure 6.  The ubiquitination-dependent LLPS of HAX1 correlates with P-bodies formation.**

(A) HAX1 domains aligned with the results of PONDR and lupred3 analyses. (B) Summary of the phase separation behaviors of purified recombinant HAX1 shown in (Appendix Fig. S1A). (C) Fusion of two mCherry-HAX1 puncta in HEK293T cells transfected with mCherry-HAX1 constructs. The two separate droplets are indicated as a, b. Scale bar, 2.5 μm. (D, E) FRAP analysis of mCherry-HAX1 droplets in HEK293T cells. Scale bar, 5 μm. (F, G) *HAX1* KO HEK293T cells transfected with vector, HAX1-mCherry (full length, FL) or IDR deletion mutants were stained with LSM14A antibodies. Representative images are shown in (F). Scale bar, 5 μm. The number of HAX1 granules within each cell was statistically analyzed (G) ($n = 50$). Error bars indicate SEM. ****$P < 0.0001$ (Student's t test). (H–K) *HAX1* KO HEK293T cells transfected with HAX1$^{WT}$-mCherry or HAX1$^{K131R}$-mCherry were treated with or without oligomycin (H, I) or co-transfected with vector or TRIM23 (J, K), and stained with LSM14A antibodies. Representative images are shown in (H, J). Scale bar, 5 μm. The number of HAX1 granules within each cell was statistically analyzed (I, K) ($n = 50$). Error bars indicate SEM. ns: no significance, ****$P < 0.0001$ (two-way ANOVA). (L–O) FRAP analysis of HAX1-mCherry in *UBB* knockdown (L), Ub overexpression (M), *TRIM23* knockout (N), or TRIM23 overexpression (O) HEK293T cells. Representative images are shown in Appendix Fig. S1G–J. The quantification of the mobile fraction at the indicated times is plotted ($n = 3$ experiments). Error bars indicate SEM. *$P < 0.05$, ****$P < 0.0001$ (two-way ANOVA). Source data are available online for this figure.

with 10% fetal bovine serum (HyClone, 10270-106) and 1% penicillin–streptomycin (Invitrogen™, 15140122), and the cells were grown in a humidified incubator under 5% $CO_2$ at 37 °C.

## Heat shock and drug treatments

For heat shock, cells were transferred to a 42 °C humidified incubator with 5% $CO_2$ for 1 h. Chemicals dissolved in DMSO were prepared and added to cells as follows: sodium arsenite (0.5 mM, Millipore Sigma 1062771000), TAK-243 (1 μM, ChemieTek CT-M7243), MG132 (4 μM, MCE, 133407-82-6), oligomycin (0.5 μM, MCE, 1404-19-9), PYR-41 (10 μM, MCE, 418805-02-4), and RA190 (100 μM, MCE, 1617495-03-0). Chemicals dissolved in ultrapure water were prepared and added to the cells as follows: 2-deoxy-D-glucose (2 mM, MCE, 154-17-6), NADH (1 mM, MCE, 606-68-8) and 1,6-HD (3%, Sigma, 240117). CoQ10 (MCE, 303-98-0) was dissolved in DMF (MCE, 68-12-2) and added to the cells at 500 μM. ATPsome (10 μM ATP, Encapsula NanoScience) was added to the medium at the beginning of heat shock or other treatments.

## Virus production and stable cell lines

For the production of lentivirus or retrovirus, HEK293T cells were co-transfected with each viral plasmid together with helper plasmids (PAX2 plus VSVG for lentivirus and pcl-10A1 for retrovirus). The virus-containing medium was collected three times at 36 h, 48 h, and 60 h after transfection and centrifuged at $300 \times g$ for 5 min. Cells were infected with the concentrated viral particles in the presence of 8 μg/ml polybrene (Sigma, H9268) and selected with the appropriate antibiotics for 3 days.

## CRISPR-mediated knockout cells

sgRNAs were designed through CRISPick (https://portals.broadinstitute.org/gppx/crispick/public), and synthesized DNA oligos were ligated into BsmBI-digested lentiCRISPR v2 vectors (Addgene, 49535). RKO cells were infected with sgRNA virus. HEK293T cells were transiently transfected with the sgRNA vector for 2 days, followed by the addition of 2 mg/ml puromycin for 3 days. For most genes (*TRIM23, FBXL7, VHL, DCAF1, DCAF7, HERC1, HECW1, KCTD16, LNX1, UBE3A,* and *LSM14A*) knock-outs, a mixture of puromycin-selected cells was used directly without single-cell isolation. For *HAX1* knockouts in HEK293T cells, monoclonal cell was obtained by single-cell

dilution in 96-well plates, and successful knockout was verified by Sanger sequencing.

## ATP measurement

The intracellular ATP level was measured by a CellTiter-Glo 2.0 assay kit (Promega G9242) according to the manufacturer's instructions.

## E3 ligase CRISPR screen

The E3 ligase sgRNA library containing 2670 sgRNAs was packaged into lentivirus, and the HEK293T cells with a moderate LSM14A-GFP fluorescence intensity were infected at an MOI < 0.2. After 3 days of puromycin screening, the first ten percent of cells with high GFP expression were sorted by flow cytometry after oligomycin treatment for 2 h. The abundance of individual sgRNAs in both groups of cells was measured using next-generation sequencing techniques. The screening hit identification was performed by MAGeCK program.

## Immunofluorescence

Cells were grown in glass-bottomed cell culture dishes (NEST, 801002), fixed with 4% paraformaldehyde (Beyotime, P0099) for 10–20 min, permeabilized with 0.5% Triton X-100 in PBS for 20 min, and blocked with 1% bovine serum albumin (BSA; Sangon Biotech, A600903-0005) for 1 h. The samples were further incubated with primary antibodies against the following targets in blocking buffer at 4 °C overnight: DCP1A, LSM14A, G3BP2, TIA1, TRIM23, DDX6, or HAX1, and then incubated with host-specific Alexa Fluor 488/594 secondary antibodies for 1 h at room temperature. The samples were washed three times with PBS and incubated with 4',6-diamidino-2-phenylindole (DAPI; Thermo, D1306) for 5 min at room temperature before microscopic imaging. Images were captured using an Olympus confocal microscope with a 60× oil objective.

## Immunoblotting, SUnSET, and co-IP assay

For immunoblotting, cells were lysed in lysis buffer (50 mM Tris-Cl at pH 7.4, 150 mM NaCl, 1% Triton X-100, 1 mM DTT), supplemented with protease inhibitor cocktail (Sangon Biotech, C600380-0001). Cell lysates were resolved using western blotting.

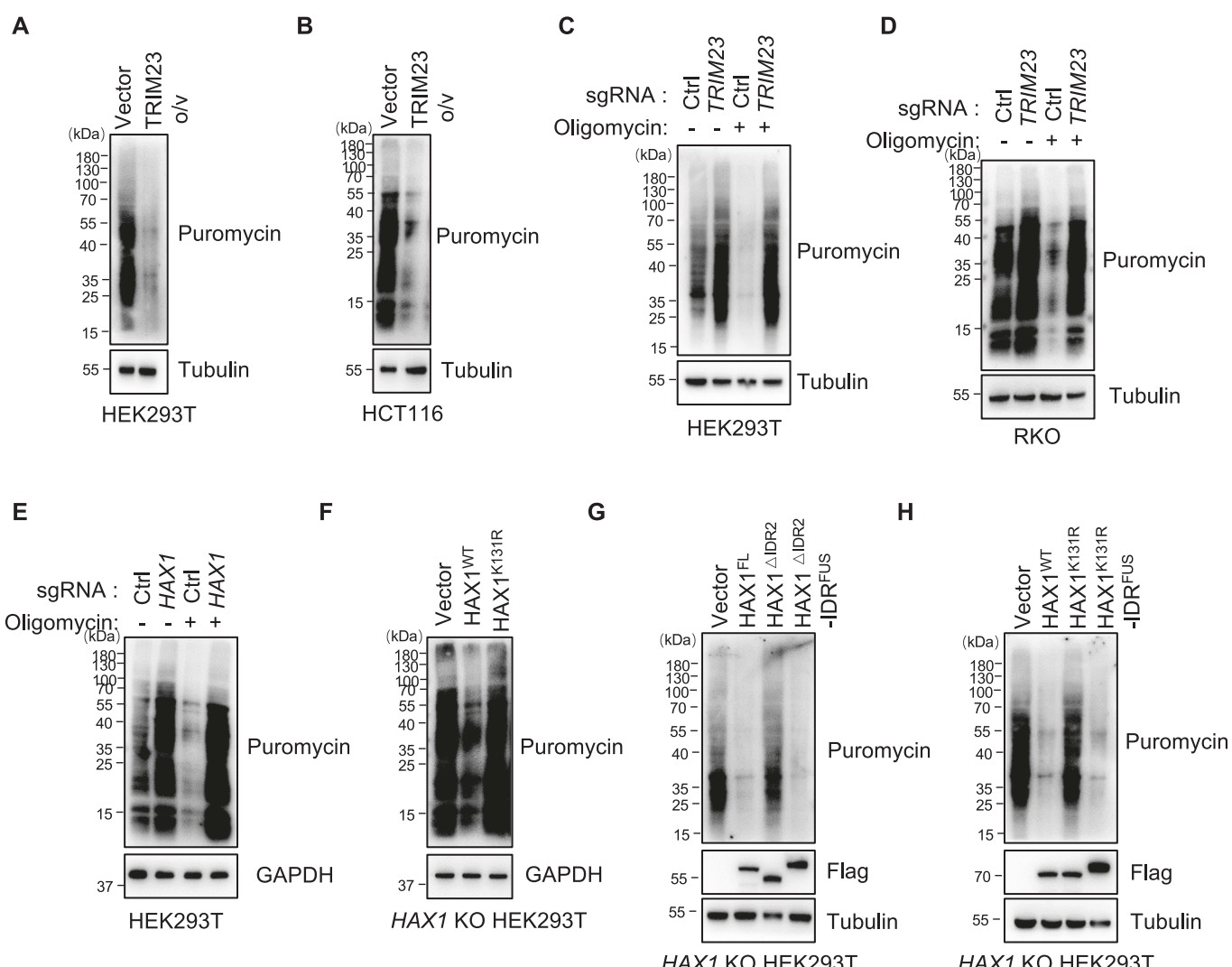

**Figure 7. TRIM23/HAX1 reduces newly synthesized proteins.**

Immunoblot of indicated cells exposed to puromycin (10 μg/ml, 30 min). HEK293T cells transfected with vector or TRIM23 (A), control and TRIM23-overexpressing (TRIM23 o/v) HCT116 cells (B), control and *TRIM23* KO HEK293T cells treated with or without oligomycin (1 h) (C), control and *TRIM23* KO RKO cells treated with or without oligomycin (1 h) (D), control and *HAX1* KO HEK293T cells treated with or without oligomycin (1 h) (E), *HAX1* KO HEK293T cells transfected with vector, HAX1^WT or HAX1^K131R (F), or *HAX1* KO HEK293T cells transfected with Flag-HAX1^ΔIDR2 or Flag-HAX1^ΔIDR2-IDR^FUS (G), or with Flag-HAX1^K131R or Flag-HAX1^K131R-IDR^FUS (H) were analyzed for newly synthesized proteins containing puromycin. Source data are available online for this figure.

For SUnSET assay, different cells with the same number were spread into the six-well plate and treated with 10 μg/ml puromycin for 30 min. Cells were collected for lysis and global protein levels were detected by western blotting. For co-IP, cell lysates were immunoprecipitated using the indicated antibodies or IgG control. Immunoprecipitates were washed three times with lysis buffer and analyzed via western blotting.

## Fluorescence recovery after photobleaching (FRAP) and live cell imaging

FRAP experiments were performed with an LSM980 confocal microscope (Zeiss) using Zeiss ZEN 3.0 (blue edition) software. Imaging was performed using a 63x Plan Apo 1.40NA oil objective, and Perfect Focus (ZEISS) was used for the duration

of the capture. For imaging, time lapses were taken using 488-nm or 561-nm imaging lasers set at 99 power and 120 ms of exposure at the maximum speed (~0.27 ms period) for 200 frames. The data were repeated in triplicate for each condition, with each replicate containing at least 3 cells. The data were analyzed with ZEISS ZEN 3.0 (blue edition). ROIs were generated in the photobleach region, a nonphotobleached cell was used, the background for each time lapse was used, and the mean intensity of each was extracted. These values were exported into Excel, where photo-bleaching and background correction were performed, and fit FRAP curves were generated. For live cell imaging, the HEK293T cells stably expressing mCherry-HAX1 were cultured at 37 °C with 5% $CO_2$ and photographed every 30 s by LSM980 confocal microscope (Zeiss) using Zeiss ZEN 3.0 (blue edition) software.

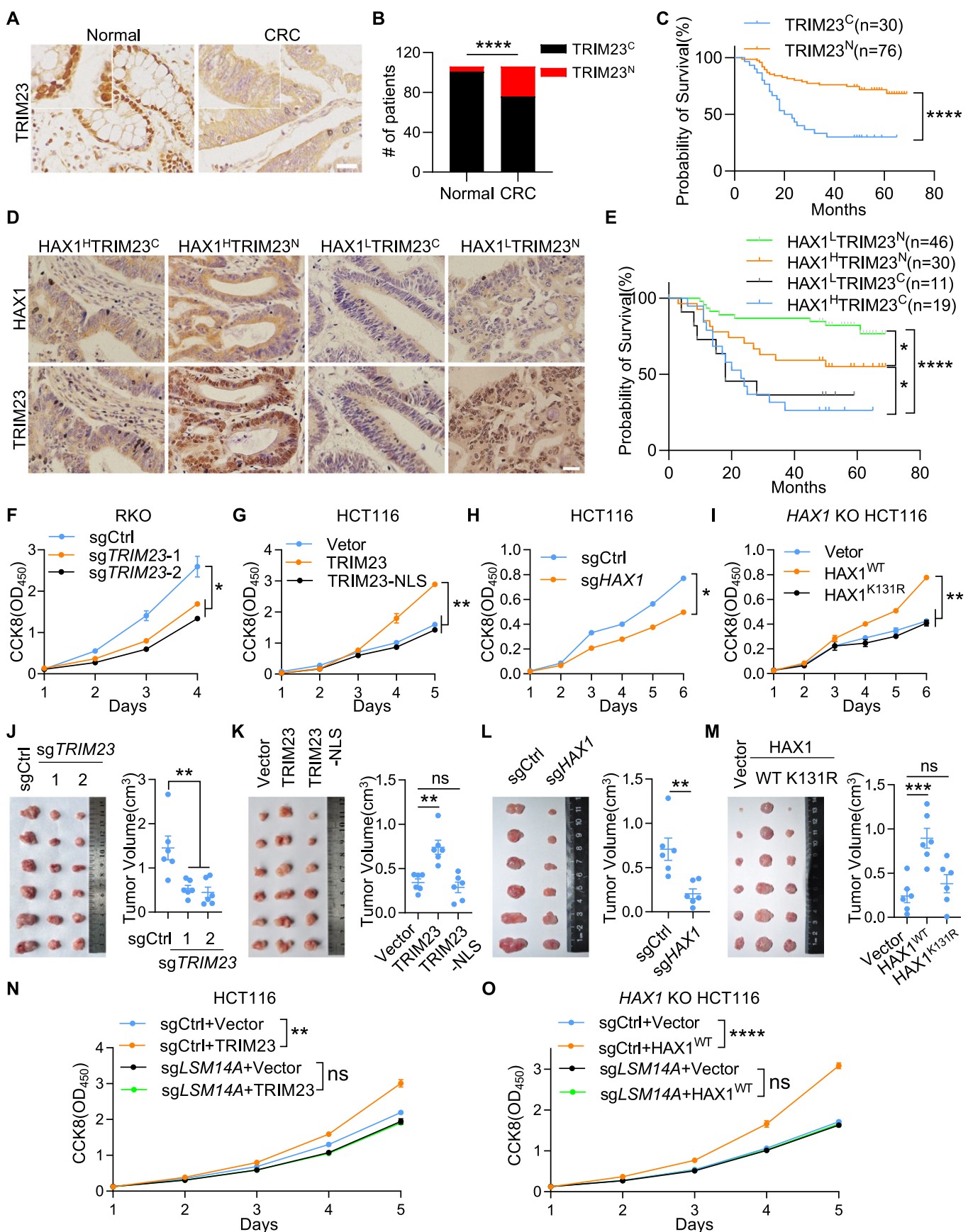

**Figure 8. Ubiquitinated HAX1 is critical for the tumorigenicity of CRC.**

(A) Immunohistochemical staining of TRIM23 in normal adjacent tissues and CRC tissues. Scale bars, 25 µm (inset, 12.5 µm). (B) Quantification of the localization of TRIM23 in (A). Superscript C, cytoplasmic location; Superscript N, nuclear location. ****$P < 0.0001$ (Chi-Squared test). (C) Kaplan–Meier survival analysis of the CRC patients in (A). ****$P < 0.0001$ (Log-rank test). (D) Immunohistochemical staining of TRIM23 and HAX1 in CRC tissues (Superscript H, high; Superscript L, low). Scale bars, 25 µm. (E) Kaplan–Meier survival analysis of the CRC patients in (D). *$P < 0.05$, ****$P < 0.0001$ (Log-rank test). (F–I) The proliferation of control or *TRIM23* knockout RKO cells (F), HCT116 cells transduced with lentivirus expressing vector, TRIM23 or TRIM23-NLS (G), control and *HAX1* knockout HCT116 cells (H), or *HAX1* KO HCT116 cells transduced with lentivirus expressing control, HAX1[WT], or HAX1[K131R] (I) were determined by CCK8 assay. At least two biological replicates are plotted as the mean ± SD. *$P < 0.05$, **$P < 0.01$ (Student's t test). (J–M) Xenograft tumor formation by the indicated CRC cells in mice. Tumor images (left) and tumor volume (right) are shown ($n = 6$ animals). Error bars indicate SEM. ns: no significance, **$P < 0.01$, ***$P < 0.001$ (Student's t test). (N, O) Control or TRIM23-overexpressing HCT116 cells (N) or *HAX1* KO HCT116 cells expressing control or HAX1 (O) were transduced with control or *LSM14A* sgRNA. The proliferation of indicated cells was determined by CCK8 assay. Three biological replicates are plotted as the mean ± SD. ns: no significance, **$P < 0.01$, ****$P < 0.0001$ (two-way ANOVA). Source data are available online for this figure.

## In vivo ubiquitination assay

Ubiquitination was detected using d-IP or IP. For d-IP, the cells were lysed in SDS-denaturing buffer (62.5 mM Tris-HCl (pH 6.8), 2% SDS, 10% glycerol, and 1.5% β-mercaptoethanol) and boiled for 10 min. The cell lysates were then diluted tenfold to fortyfold in native lysis buffer (50 mM Tris-HCl (pH 7.4), 0.5% Triton X-100, 200 mM NaCl, and 10% glycerol). For IP, lysates were prepared as in immunoprecipitation assay. After centrifugation at 13,000 r.p.m. for 5 min, the supernatants were immunoprecipitated using anti-Flag/HAX1 antibodies at 4 °C for 4 h or overnight. Then, the immunocomplexes were mixed with the washed magnetic beads at 4 °C for 4 h or overnight. The immunocomplexes were washed three times with native lysis buffer and resolved via western blotting.

## Preparation of soluble (S2) and particular (P2) fractions

As described previously (Rzeczkowski et al, 2011; Teixeira et al, 2005), two million HEK293T cells subjected to different treatments were washed in ice-cold PBS and harvested by centrifugation at $2,00 \times g$ for 5 min. The cell pellet was then lysed in 100 µl of lysis buffer (50 mM Tris at pH 7.6, 50 mM NaCl, 5 mM $MgCl_2$, 0.1% NP-40, 1 mM β-mercaptoethanol, 1× protease inhibitor complete mini EDTA free, 0.4 U/µL RNase inhibitor) on ice for 15 min with vortexing every 5 min. The lysates were centrifuged at $2000 \times g$ for 2 min, after which the supernatant was recovered. Ten microliters of supernatant were taken as the whole-cell lysate (WCL), and the remaining 90 µL of supernatant was centrifuged again at $10,000 \times g$ for 10 min at 4 °C. Most of the supernatant was taken as the soluble fraction (S2), and the pellet was resuspended in 90 µl of lysis buffer as the P body–enriched particular fraction (P2). Equal volumes of the pellet and supernatant fractions were used for Western blot analysis.

## Liquid–liquid phase separation

In vitro LLPS experiments were performed at room temperature. LLPS of recombinant HAX1 (Proteintech, Ag27244) was induced by the addition of the indicated concentrations of PEG. The samples were mixed in low binding tubes and transferred to a sandwiched chamber created by cover glass and a glass slide with a double-sided spacer. The samples were observed under a DIC microscope using a Leica DMi8 microscope with a 20× objective. All images were captured within 5 min after LLPS induction.

## Cell viability and colony formation assays

Cell proliferation and viability were detected by CCK8 assay. For the CCK-8 assays, cells ($1 \times 10^3$ cells per well) were seeded in 96-well plates in complete RPMI 1640 medium. Beginning on the second day, 10 µl of CCK8 reagent (DOJINDO, CK04) was added to each well and incubated at 37 °C for an additional 2 h, after which the absorbance was measured at 450 nm. All the experimental data were obtained from three replicates. The results were graphically analyzed using GraphPad Prism 9 (GraphPad Software). For the colony formation assay, cells (500 cells per well) were seeded in six-well plates in complete RPMI 1640 medium. The cells were fixed with 4% paraformaldehyde for 15 min and stained with crystal violet (Beyotime, C0121) at room temperature for 10 min.

## In vivo xenograft tumor assay

The animal experiments were approved by the animal ethics committee of Southern Medical University and carried out in accordance with its guidelines. All the mice were housed in specific pathogen-free condition. Male athymic nude mice (4–6 weeks old) were obtained from the Animal Experimental Center of Southern Medical University. To establish CRC xenografts, ~$1 \times 10^6$ tumor cells in 0.1 mL of phosphate-buffered saline (PBS) were injected subcutaneously into the right flank of the nude mice. When palpable tumors were observed (~3 weeks), the mice were euthanized, and the tumors were removed.

## Immunohistochemistry (IHC)

The human CRC tissue array was prepared by Professor Yongjian Deng from the Department of Pathology, Southern Medical University. The CRC tissue array was incubated at 58 °C for 2 h, after which the samples were deparaffinized with xylene and rehydrated through an ethanol gradient. For the immunohistochemical staining of TRIM23 and HAX1, antigen retrieval was performed by heating at 95 °C in citrate buffer (Zsbio, ZLI-9065) for 30 min and cooling to room temperature. The sections were incubated in 3% hydrogen peroxide (ZSBio, PV-6000) for 10 min and blocked in goat serum (ZSBio, ZLI-9021) for 1 h. The sections were incubated with TRIM23 (Sigma, 1:200) or HAX1 (Proteintech, 1:200) at 4 °C overnight, and then incubated with an enzyme-labeled goat anti-rabbit IgG polymer (ZSBio, PV-6000) at room temperature for 1 h. After DAB staining, the sections were dehydrated with an ethanol gradient and xylene, air-dried and

sealed with neutral gum (ZSBio, ZLI-9555). The tissues were visualized and imaged using an inverted microscope (Olympus, Japan). Immunohistochemical staining intensity scores were expressed as negative (1), weak (2), medium (3), or strong (4). The staining of 500 cells was observed, and the final score was determined by the sum of the different staining intensities multiplied by the number of cells.

### Quantification and statistical analysis

Images were analyzed with Image J. Statistical parameters were all reported in Figure legends. Quantification was performed using GraphPad Prism 9 (GraphPad Software). Student's t test was used to compare two groups of data, and two-way ANOVA was used to compare multiple groups of data. $P$ values are indicated in the figure legends. $P < 0.05$ was considered to indicate statistical significance.

## Data availability

All the data are available in the manuscript, Figs, supplemental information and the Source data files. This study includes no data deposited in external repositories.

The source data of this paper are collected in the following database record: biostudies:S-SCDT-10_1038-S44318-024-00120-6.

## Peer review information

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

## Acknowledgements

We thank Dr. Jingqian Huang and Yikun Yao for editorial assistance, Dr Yongjian Deng for the CRC tissue array and Fang Yao for assistance with microscopy. This research was supported by grants from the National Natural Science Foundation of China (NSFC) (No. 82071742, 32270926, and 81901578) and the Guangzhou Municipal Science and Technology Bureau (No. 202102020162) to ZW; NSFC (No. 82272192 and 82281240019); the Program of Shanghai Academic Research Leader (No. 22XD1420400) to WC.

## Author contributions

**Wanqi Zhan**: Data curation; Formal analysis; Validation; Writing—original draft. **Zhiyang Li**: Data curation; Formal analysis; Writing—original draft; Writing—review and editing. **Jie Zhang**: Data curation; Formal analysis. **Yongfeng Liu**: Resources; Validation. **Guanglong Liu**: Resources; Formal analysis. **Bingsong Li**: Resources. **Rong Shen**: Resources; Formal analysis. **Yi Jiang**: Resources. **Wanjing Shang**: Resources; Writing—review and editing. **Shenjia Gao**: Resources. **Han Wu**: Resources. **Ya'nan Wang**: Resources. **Wankun Chen**: Data curation; Formal analysis. **Zhizhang Wang**: Conceptualization; Supervision; Writing—original draft; Writing—review and editing.

Source data underlying figure panels in this paper may have individual authorship assigned. Where available, figure panel/source data authorship is listed in the following database record: biostudies:S-SCDT-10_1038-S44318-024-00120-6.

## Disclosure and competing interests statement

The authors declare no competing interests.

# Expanded View Figures

**Figure EV1.  ATP regulates P-bodies formation.**

(**A**) The cellular ATP content of HEK293T cells treated with 2-DG or/and oligomycin was detected. Three biological replicates are plotted as the mean ± SD. ***$P < 0.001$ (Student's t test). (**B, C**) HEK293T cells treated with 2-DG or/and oligomycin (2 h) were exposed to sodium arsenite (2 h) and immunolabeled for TIA1 and G3BP2. Representative images are shown in (**B**). The number of stress granules represented by TIA1/G3BP2 within each cell is plotted in (**C**) ($n = 50$). Scale bar, 5 μm. Error bars indicate SEM. ns: no significance, ****$P < 0.0001$ (two-way ANOVA). (**D, E**) RKO cells were treated with oligomycin (1 h) and stained for LSM14A and DCP1A. Representative images are shown in (**D**). The number of P-bodies within each cell is plotted in (**E**) ($n = 50$). Scale bar, 5 μm. Error bars indicate SEM. ****$P < 0.0001$ (Student's t test). (**F**) The number of P-bodies of different sizes in Fig. 1A. ImageJ software was used to determine the particle size of the P-bodies (large, diameter >0.2 μm; small, diameter <0.2 μm). Error bars indicate SD. Red and black lines are used for the comparison of large and small P-bodies. ****$P < 0.0001$ (two-way ANOVA). (**G**) The cellular ATP content of HEK293T cells treated with or without NADH plus CoQ10 (2 h) and exposed to heat shock (42 °C) (1 h). Three biological replicates are plotted as the mean ± SD. ****$P < 0.0001$ (two-way ANOVA). (**H, I**) RKO cells were treated with or without NADH plus CoQ10 (2 h) and stained for LSM14A and DCP1A. Representative images are shown in (**H**). The number of P-bodies within each cell is plotted in (**I**) ($n = 50$). Scale bar, 5 μm. Error bars indicate SEM. ****$P < 0.0001$ (Student's t test). (**J**) ATP assay of HEK293T cells exposed to heat shock (42 °C) (1 h) or treated with oligomycin (2 h) in the presence of liposome or ATPosome. Three biological replicates are plotted as the mean ± SD. ****$P < 0.0001$ (two-way ANOVA). (**K, L**) The number of P-bodies of different sizes in Fig. 1E (**K**) and Fig. 1G (**L**). Error bars indicate SD. ns: no significance, ***$P < 0.001$, ****$P < 0.0001$ (two-way ANOVA). (**M, N**) RKO cells were treated with liposome or ATPsome (1 h) in the presence or absence of oligomycin (2 h) and stained for LSM14A and DCP1A. The cells were imaged (**M**), and the number of P-bodies within each cell was quantified (**N**) ($n = 50$). Scale bar, 5 μm. Error bars indicate SEM. *$P < 0.05$, ****$P < 0.0001$ (two-way ANOVA).

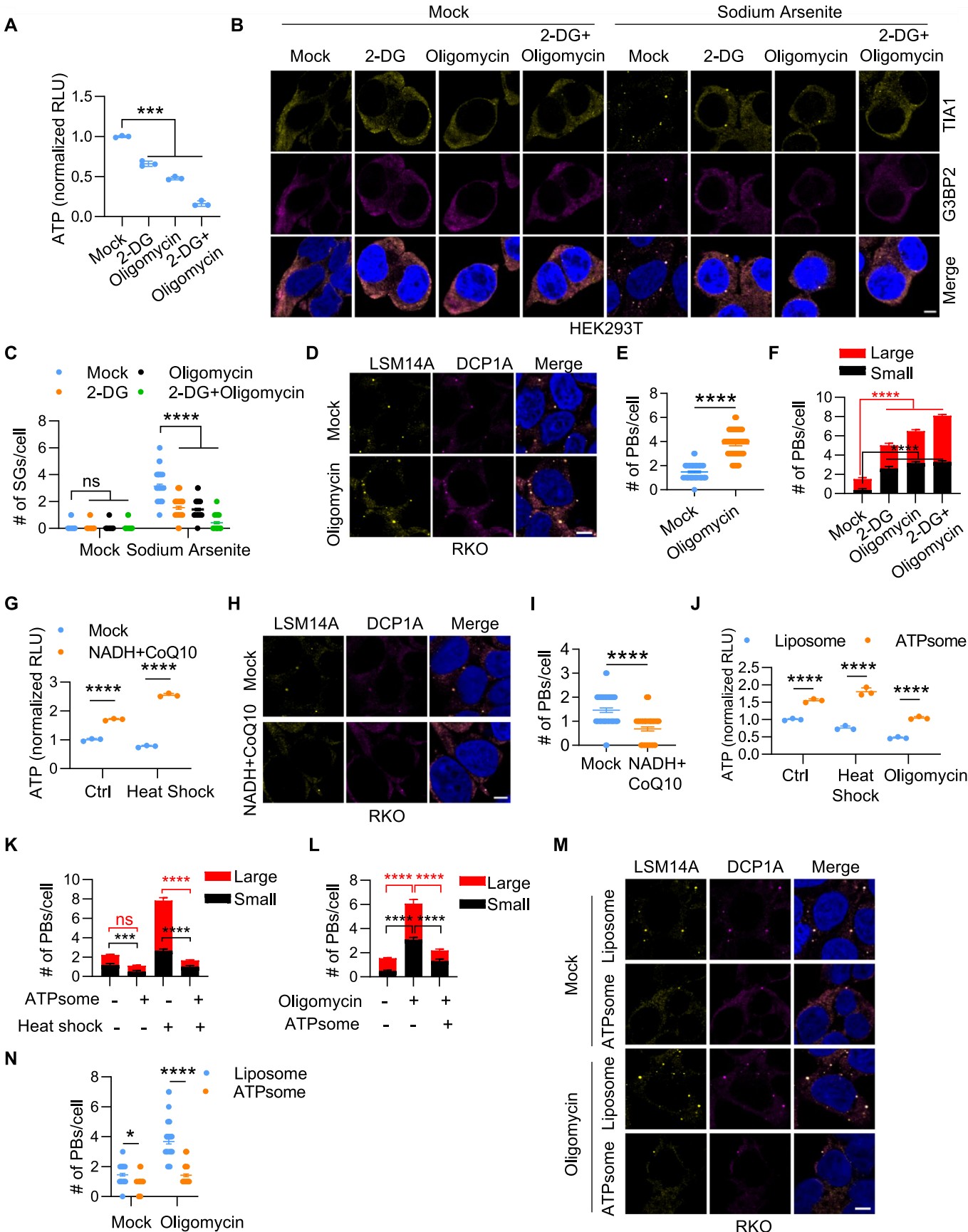

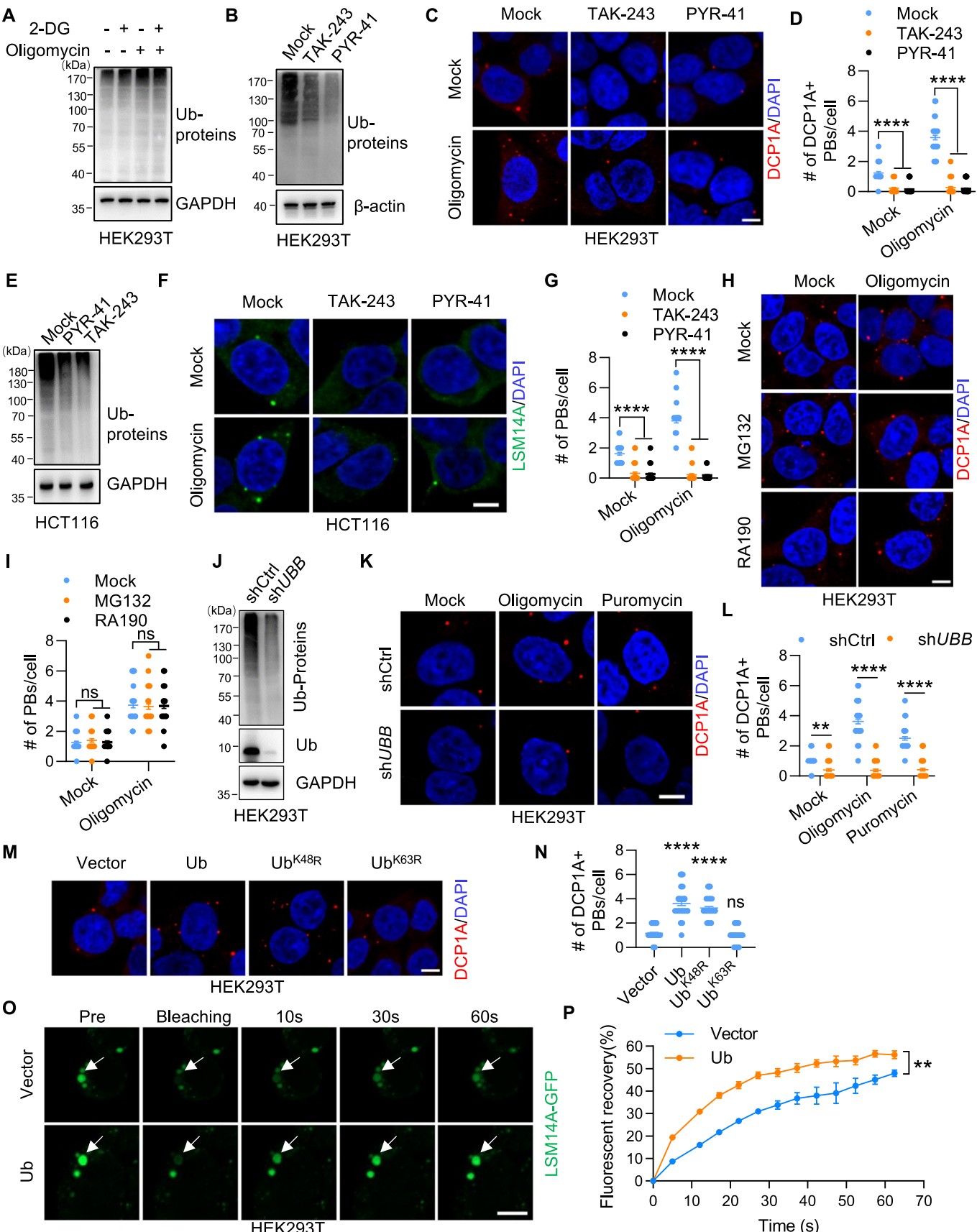

**Figure EV2.  Ubiquitin regulates oligomycin-induced P-bodies dynamics.**

(A) HEK293T cells were treated with 2-DG or/and oligomycin for 2 h. The ubiquitination level of the proteins was detected by immunoblotting. (B) Immunoblot of HEK293T cells treated with TAK-243 or PYR-41 (2 h). (C, D) HEK293T cells were treated with TAK-243 or PYR-41 (1 h) in the presence or absence of oligomycin (1 h) and immunolabeled for DCP1A. Representative images are shown in (C). Scale bar, 5 μm. The number of P-bodies within each cell is plotted in (D) ($n = 50$). Error bars indicate SEM. ****$P < 0.0001$ (two-way ANOVA). (E) Immunoblot of HCT116 cells treated with TAK-243 or PYR-41 (2 h). (F, G) HCT116 cells were treated with TAK-243 or PYR-41 in the presence or absence of oligomycin (1 h) and immunolabeled for LSM14A. Representative images are shown in (F). Scale bar, 5 μm. The number of P-bodies within each cell is plotted in (G) ($n = 50$). Error bars indicate SEM. ****$P < 0.0001$ (two-way ANOVA). (H, I) HEK293T cells were treated with MG132 or RA190 (1 h) in the presence or absence of oligomycin (1 h) and immunolabeled for DCP1A. Representative images are shown in (H). Scale bar, 5 μm. The number of P-bodies within each cell is plotted in (I). Error bars indicate SEM. ns: no significance (two-way ANOVA). (J) Immunoblot of control and *UBB*-knockdown HEK293T cells. (K, L) Control and *UBB*-knockdown HEK293T cells were treated with oligomycin (1 h) or puromycin (1 h), and stained for DCP1A and DAPI. Representative images are shown in (K). The number of P-bodies within each cell is plotted in (L) ($n = 50$). Error bars indicate SEM. **$P < 0.01$, ****$P < 0.0001$ (two-way ANOVA). (M, N) HEK293T cells transfected with vector, Ub, Ub^K48R or Ub^K63R were stained for DCP1A and DAPI. Representative images are shown in (M). The number of P-bodies within each cell is plotted in (N) ($n = 50$). Scale bar, 5 μm. Error bars indicate SEM. Ns: no significance, ****$P < 0.0001$ (Student's t test). (O, P) FRAP analysis of GFP-LSM14A in HEK293T cells transfected with vector or Ub. White arrows indicate the target droplets for FRAP. Scale bar, 5 μm. $n = 3$, Error bars indicate SEM. **$P < 0.01$ (two-way ANOVA).

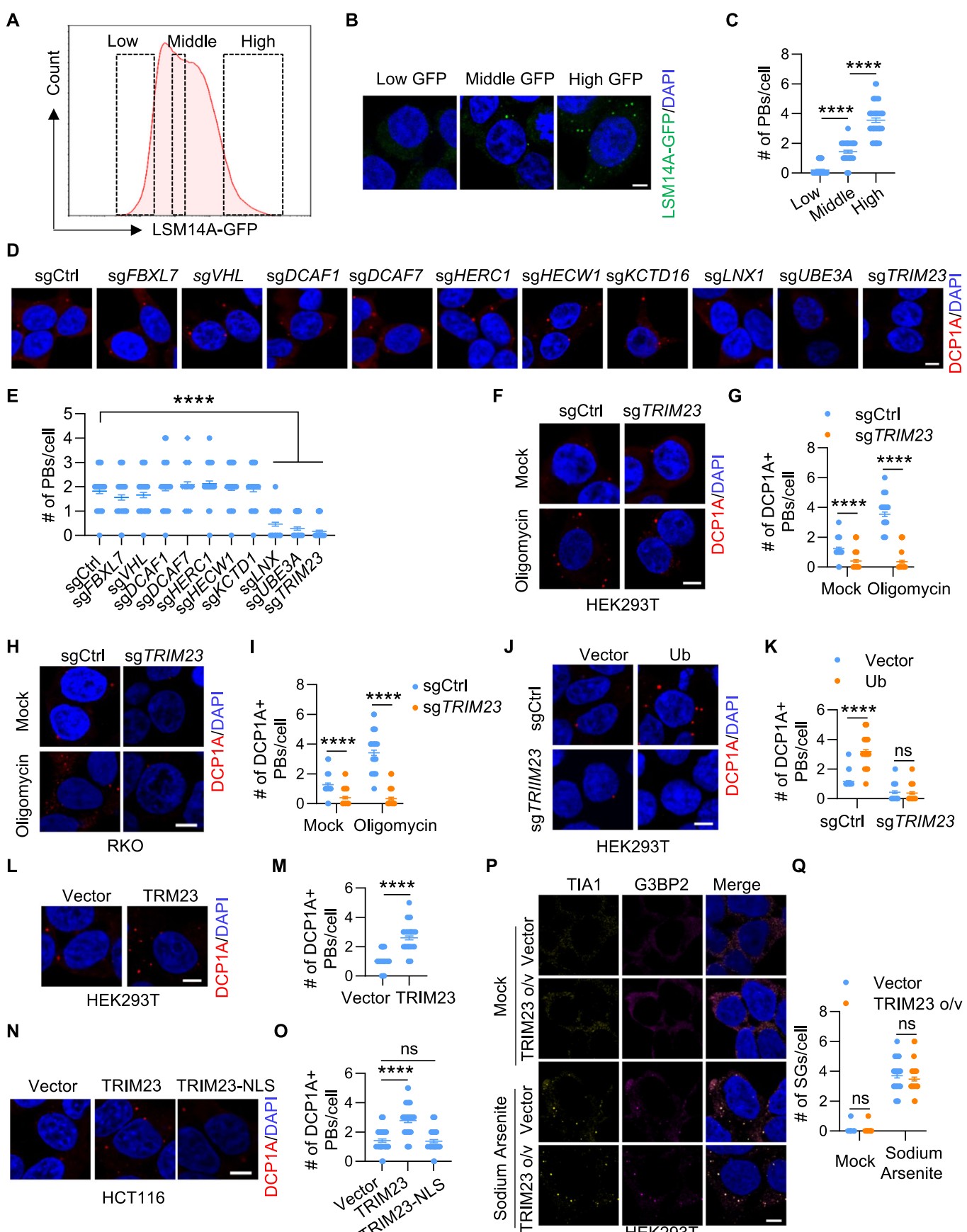

**Figure EV3. TRIM23 promotes P-bodies formation.**

(A–C) HEK293T cells expressing GFP-LSM14A with low, middle, and high fluorescence intensities were isolated by fluorescence-activated cell sorting (A). Cells were imaged after sorting (B), and the number of P-bodies within each cell from one of two biological replicates is plotted in (C) ($n = 50$). Scale bar, 5 µm. Error bars indicate SEM. ****$P < 0.0001$ (Student's t test). (D, E) HEK293T cells were transfected with LentiCRISPR constructs targeting the indicated genes for 48 h and selected with puromycin for 72 h. Cells were stained for DCP1A and DAPI and imaged (D). Scale bar, 5 µm. The number of P-bodies within each cell is plotted in (E) ($n = 50$). Error bars indicate SEM. ****$P < 0.0001$ (Student's t test). (F–K) *TRIM23* KO HEK293T cells (F, G, J, K) or *TRIM23* KO RKO cells (H, I) were treated with or without oligomycin (1 h) (F–I) or transfected with vector or Ub (J, K) and stained for DCP1A and DAPI. Representative images are shown in (F, H, J). The number of P-bodies within each cell is plotted in (G, I, K) ($n = 50$). Scale bar, 5 µm. Error bars indicate SEM. ns: no significance, ****$P < 0.0001$ (two-way ANOVA). (L–O) HEK293T cells transfected with TRIM23 or vector and HCT116 cells transduced with lentivirus expressing control, TRIM23 or TRIM23-NLS were stained for DCP1A and DAPI. Representative images are shown in (L, N). The number of P-bodies within each cell is plotted in (M, O) ($n = 50$). Scale bar, 5 µm. Error bars indicate SEM. ns: no significance, ****$P < 0.0001$ (Student's t test). (P, Q) HEK293T cells transfected with TRIM23 or vector were exposed to sodium arsenite (2 h) and immunolabeled for TIA1 and G3BP2. Representative images are shown in (P). The number of stress granules within each cell is plotted in (Q) ($n = 50$). Scale bar, 5 µm. Error bars indicate SEM. ns: no significance (two-way ANOVA).

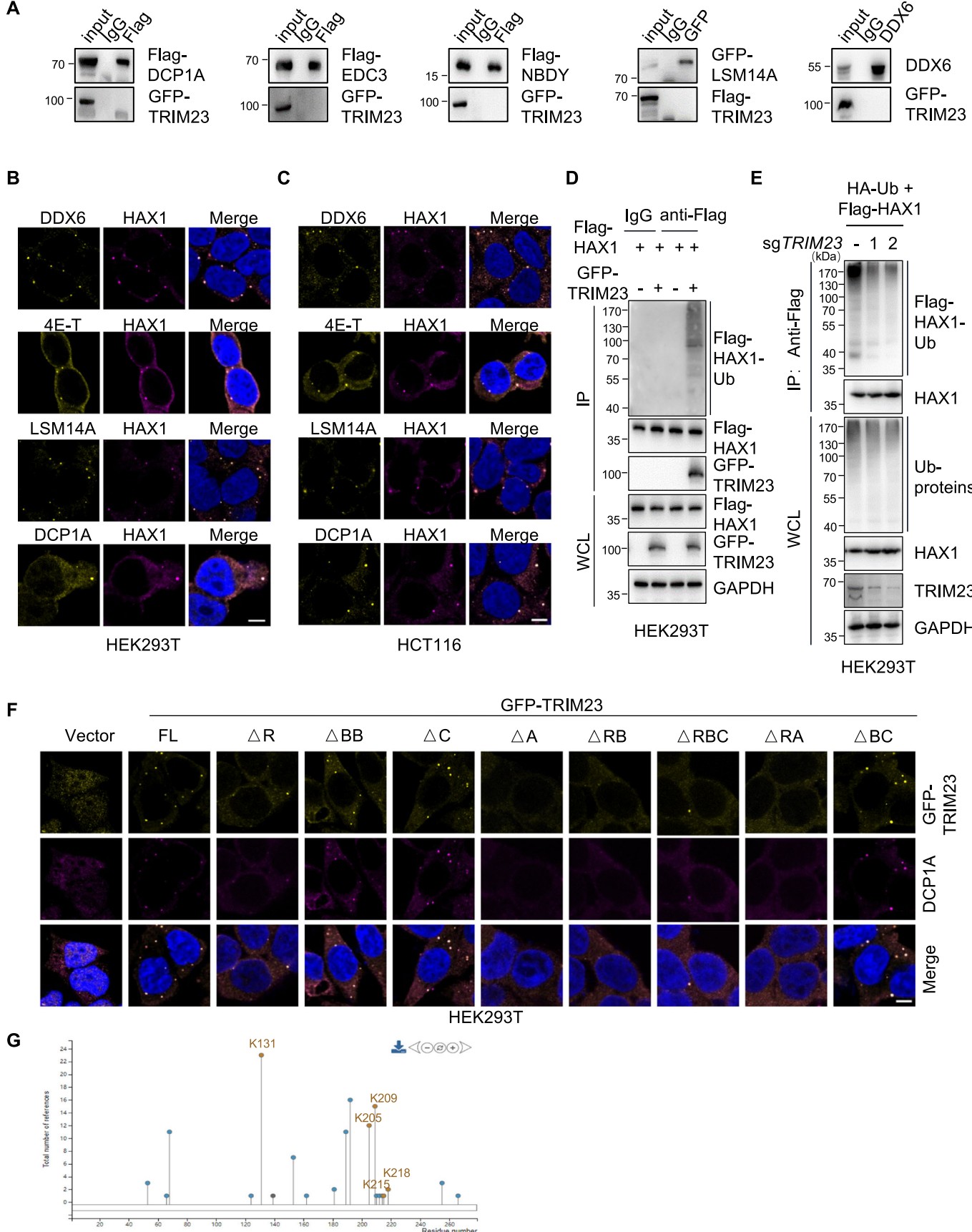

◄    **Figure EV4.   TRIM23 interacts with HAX1 to catalyze ubiquitination.**

(**A**) HEK293T cells were transfected with GFP-TRIM23 along with flag-DCP1A, flag-EDC3 or flag-NBDY, or GFP-LSM14A with flag-TRIM23, or GFP-TRIM23 alone. Cell lysates were immunoprecipitated using control IgG or the indicated antibodies. IP samples and whole-cell lysates (input) were analyzed via western blotting. (**B**, **C**) Immunofluorescence staining of HEK293T cells (**B**) or HCT116 cells (**C**) for endogenous HAX1 and DDX6, 4E-T, LSM14A, or DCP1A. Scale bar, 5 μm. (**D**) HEK293T cells were transfected with Flag-HAX and GFP-TRIM23. Cell lysates were immunoprecipitated using IgG or anti-Flag antibodies. IP samples and whole-cell lysates (WCLs) were analyzed via western blotting. (**E**) Control and *TRIM23*-knockout HEK293T cells were transfected with HA-Ub and Flag-HAX1. Cell lysates were immunoprecipitated using anti-Flag antibodies. IP samples and WCLs were analyzed using western blotting. (**F**) HEK293T cells were transfected with vector-GFP, TRIM23-GFP-FL or its deletion mutants and stained for DCP1A and DAPI. Scale bar, 5 μm. (**G**) Ubiquitination sites (Lys131/205/209/215/218) of HAX1 reported on PhosphoSitePlus.

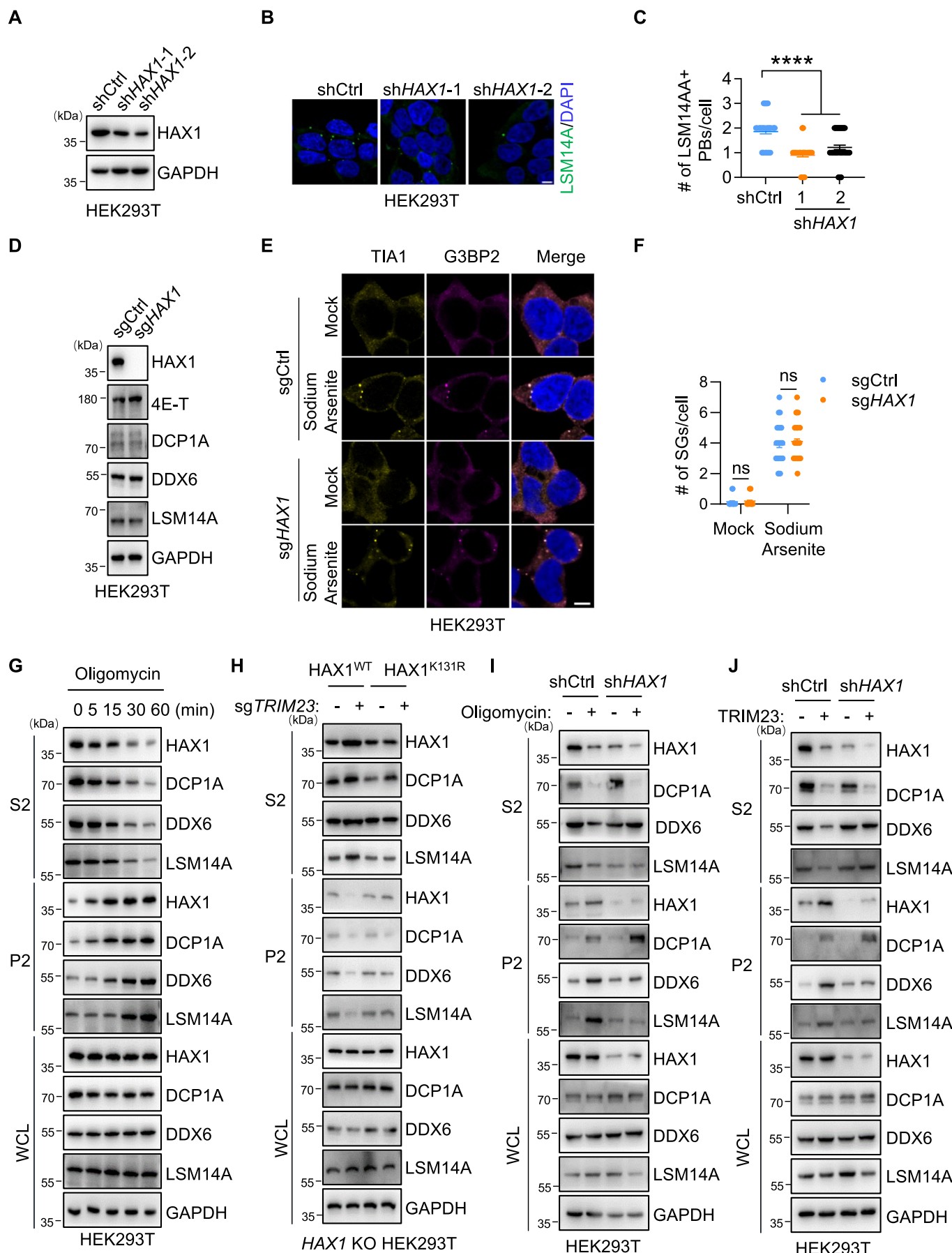

◀   **Figure EV5.   The condensation of P-bodies induced by energy stress depends on ubiquitinated HAX1.**

(A) Immunoblot of HEK293T cells transfected with shCtrl or two independent HAX1 shRNAs. (B, C) Control and HAX1 knockdown (sh*HAX1*) HEK293T cells were stained for LSM14A and DAPI. Representative images are shown in (B). The number of P-bodies within each cell is plotted in (C) ($n = 50$). Scale bar, 5 μm. Error bars indicate SEM. ****$P < 0.0001$ (Student's t test). (D) The expression of HAX1, 4E-T, DCP1A, DDX6, and LSM14A in *HAX1* KO HEK293T cells was detected by immunoblotting. (E, F) Control and *HAX1* KO HEK293T cells were exposed to sodium arsenite (2 h) and immunolabeled for TIA1 and G3BP2 (E). Scale bar, 5 μm. The number of stress granules within each cell is plotted in (F) ($n = 50$). Statistical Error bars indicate SEM. Ns: no significance (two-way ANOVA). (G) S2 and P2 fractions from HEK293T cells treated with oligomycin for the indicated times were analyzed by immunoblotting. (H) S2 and P2 fractions from *HAX1* KO HEK293T cells transfected with HAX1[WT] or HAX1[K131R] along with sgCrtl or sg*TRIM23* were analyzed by immunoblotting. (I, J) S2 and P2 fractions from control and HAX1-knockdown (sh*HAX1*) HEK293T cells with or without oligomycin treatment (1 h) (I) or TRIM23 overexpression (J) were analyzed by immunoblotting.

                                                    