## [Peer Review File · The EMBO Journal]

Energy stress promotes P-bodies formation via lysine-63-linked polyubiquitination of HAX1

Zhizhang Wang, Wanqi Zhan, Zhiyang Li, Jie Zhang, Yongfeng Liu, Guanglong Liu, Bingsong Li, Rong Shen, Yi Jiang, Wanjing Shang, Shenjia Gao, Han Wu, Ya'nan Wang, and Wankun Chen

Corresponding author(s): Zhizhang Wang (wangzz89@smu.edu.cn) , Wankun Chen (chen.wankun@zs-hospital.sh.cn)

Review Timeline:

Submission Date:	10th Sep 23
Editorial Decision:	14th Nov 23
Revision Received:	15th Feb 24
Editorial Decision:	18th Mar 24
Revision Received:	22nd Mar 24
Accepted:	15th Apr 24

Editor: *Cornelius Schneider*

Transaction Report:

Dear Dr. Wang,

Thank you for submitting your manuscript for consideration by the EMBO Journal and for sharing a preliminary revision plan with me.

Based on the overall positive evaluation of the manuscript by the referees and your willingness to engage in a major revision as indicated during the pre-decision consultation, I would like to invite you to submit a revised version of the manuscript, addressing the comments of all three reviewers. I should add that it is EMBO Journal policy to allow only a single round of revision, and acceptance of your manuscript will therefore depend on the completeness of your responses in this revised version. If you have any additional questions or want to discuss the revisions further, I am happy to do so by email or video conferencing.

We generally allow three months as standard revision time, which can be extended to 6 months in case of major revisions, such as the experiments required here. As a matter of policy, competing manuscripts published during this period will not negatively impact on our assessment of the conceptual advance presented by your study. However, we request that you contact the editor as soon as possible upon publication of any related work, to discuss how to proceed. Should you foresee a problem in meeting the deadline, please let us know in advance and we may be able to grant an extension.

Thank you for the opportunity to consider your work for publication. I look forward to your revision.

Yours sincerely,

Cornelius Schneider

Cornelius Schneider, PhD
Editor
The EMBO Journal
c.schneider@embojournal.org

Please remember: Digital image enhancement is acceptable practice, as long as it accurately represents the original data and

conforms to community standards. If a figure has been subjected to significant electronic manipulation, this must be noted in the figure legend or in the 'Materials and Methods' section. The editors reserve the right to request original versions of figures and the original images that were used to assemble the figure.

We realize that it is difficult to revise to a specific deadline. In the interest of protecting the conceptual advance provided by the work, we recommend a revision within 3 months (12th Feb 2024). Please discuss the revision progress ahead of this time with the editor if you require more time to complete the revisions. Use the link below to submit your revision:

Referee #1:

In this paper, Zhan et al. show that ATP depletion leads to an increase in the number of P-bodies per cell. They demonstrate that this increase in P-body number depends on the ubiquitination of HAX1 at K131 by the E3 ubiquitin ligase TRIM23. Furthermore, HAX1 ubiquitination, as well as functional TRIM23, are required for in vitro condensation of HAX1. Next, the authors show that HAX1 is an essential P-body component functioning early in P-body assembly. Finally, they present evidence that TRIM23 and HAX1 are involved in colorectal cancer proliferation.

The presented work contains an impressive amount of data, many of which are valuable additions to the field. However, to make the story more accessible we suggest that the authors consider to revise the manuscript and reduce the amount of data displayed in this work.

In addition, the work provides insight into a new aspect of P-body formation and draws a connection between P-bodies and energy stress. Yet, the connection between energy depletion and heat shock is not entirely clear, considering that ATP reduction is only one of many effects of heat shock response. Furthermore, the effects on ATP reduction are marginal compared to oligomycin treatment and rescue with ATPsome and NADH+CoQ10 increases the ATP levels by 1.5 and 2.5-fold respectively. The authors need to consider that this possibly results in hydrotropic effects of ATP on condensates dissolution.

Finally, the involvement of the axis HAX1/TRIM23 in colorectal cancer is well developed but the involvement of P-bodies is merely a correlation and the authors need to discuss it as such.

In summary, this paper provides convincing evidence for a role of ubiquitination of HAX1 by TRIM23 for the formation of P-bodies. However, the results on P-body induction by energy depletion and the overall OXPHOS gene reduction in HAX1 KO and HAX1K131R are less convincing. Nevertheless, upon adequate revision the work should be published in EMBO Journal.

Major Points

- The authors repeatedly conclude that P-bodies are the responsible factors for the observed effects. This conclusion cannot be drawn with the presented evidence and need to be toned down and any implied causal connection between P-bodies and downstream effects should be revised and rephrased. For example, an importance of TRIM23/HAX1 for the tumorigenicity of CRC is supported, yet any importance of P-bodies for tumor development is not, and the author need to consider that TRIM23/HAX1 might be able to cause the observed effect independently of P-bodies. The same applies to the conclusions on phase separation. The conclusion in the title that "phase separation of HAX1 triggers P-body assembly" is based on the observation that HAX1 can phase separate in vitro and that HAX1_K131R as well as HAX1_ΔIDR2 disrupts P-body assembly in cells. However, the K131 ubiquitination could also control P-body assembly in ways unrelated to phase separation. Therefore, the presented data should not be presented as depicting a causal relationship between phase separation of HAX1 and P-body formation.
- The abstract should be carefully re-written in a logical manner avoiding unclear expressions like "unnecessary biological process restriction". It would be advisable that a native English speaker proofreads the article especially regarding logical consistency of the writing.
- "We also found that heat shock decreased cellular ATP levels, which could be reversed by supplement with ATPsome or CoQ10+NADH (Extended Data Fig. 1f, g), suggesting that low ATP levels were responsible for P-Body assembly induced by heat shock." This statement is not supported by data. The reduction of ATP in HS condition is lower compared to the 2-DG treatment which alone barely increases P-Body number. While HS reduces ATP levels it might have additional effect on PB formation independent of changes in ATP level. The authors need to comment and discuss this. Furthermore, if low ATP levels are responsible for the increase in P-body numbers, then colorectal cancer cell lines should display no P-bodies if treated with ATPsome or CoQ10+NADH but increase their number after oligomycin treatment. If this is not the case the initial general statement should be toned down as "Low ATP levels increases P-bodies number in a cell line dependent context"
- It is not clear why sometimes 2-DG+Oligomycin is used and at other times only oligomycin considering they have a different effect on reducing ATP levels. Could the author comment on this and/or clarify this in the text?
- "Because the core function of P-body is mRNA degradation, we focused on oligomycin-repressed genes". This statement is not correct as a large body of literature shows that P-bodies are not sites of mRNA decay. Furthermore, Extended Figure 5d-g is difficult to interpret and should be better described/analyzed in the text. Especially, the oligomycin treated vs untreated samples need to be clarified the corresponding fraction of genes belonging to the different groups have to be made clear. On the same

line "The most significantly downregulated genes in wild-type cells were electron transfer chain genes of OXPHO, which could not be repressed by oligomycin in HAX1 knockout and K131R mutant cells (Extended Data Fig. 5f, g)". Carefully looking at the graph and comparing the oligomycin treated lines, it seems that the mitochondrial genes are similarly repressed in HAX1 WT, HAX1 KO and HAX1K131R. Can the authors provide side by side quantifications of these genes for better comparison?

• Several references are not correct and do not refer to the points made. For example, references 42,43 do not support the following claim: "Many P-body proteins possessing LLPS ability harbor IDRs, which are considered in the context of the regulation of P-body assembly". The authors need to check all references carefully.

Minor points

- DCP1 α should be changed in DCP1A
- NaAsO₂ should be changed to sodium arsenite to be consistent with figures.
- μ M has to be changed to μ M
- One consistent form of writing "P-bodies" should be used
- Page 9 line 9: change aerobic with anaerobic
- The axis label "Time(s)" should be "Time (s)" to distinguish from the plural
- For those experiments where only technical replicates are used, it needs to be explained why.
- Can the authors comment on the observation that an increase in ATP levels does not decrease P-body numbers in steady state levels but only under heat shock?
- In Figure 2b the addition of ATPsome seems to decrease the ubiquitin levels also without the addition of oligomycin. Can the authors comment on this and discuss how this might affect their conclusions?
- Figure 5J does not include any blot for DCP1A, yet in the text it is specified that there is no binding.
- Extended data figure 1F: The control should be DMF not DMSO
- Extended data figure 1J: Statistics should be checked
- Time points for drug treatment and heat shock are missing. This information needs to be provided at least in the methods.
- Figure panels should be rearranged to fit the logical sequence of the text wherever possible.
- Given the many different cell lines used to the authors should label each figure with the corresponding cell line for better understanding.
- The statement "In contrast, forced ubiquitin expression facilitated the recovery of LSM14A after FRAP, suggesting that ubiquitination is necessary in P-body LLPS (Extended Data Fig.2m, n)" is not a valid conclusion because a faster FRAP recovery is not necessarily connected to LLPS.
- Please refrain from using green and red in a merged image.
- Can the authors comment on their claim that "PYR-41 treatment also significantly shortened the recovery time of LSM14A (Fig. 2k, l)". It looks rather that the mobile fraction is reduced.
- Figure E4C "anti-flag" is spelled wrong.
- "Taken together, our findings validate that ubiquitinated HAX1 is required for cells to respond to energy stress by controlling P-body assembly." Can the authors elaborate what "response" they refer to?
- Claiming that HAX1 is the pioneer protein for P-body assembly is premature because not all P-body proteins have been tested. You can say that is "a pioneer protein" though.

Referee #2:

In this manuscript, Zhan et al studied P-body (PB) formation and dynamics in the context of the energy stress consecutive to ATP depletion. By imaging stressed human cells, including FRAP experiments, they showed that ATP depletion induced de novo PB assembly in an ubiquitin dependant manner. By using a CRISPR screen and various knock-out, knock-down and over-expression experiments, they found that the E3 ligase TRIM23 was involved in response to ATP stress through HAX1 ubiquitination. The mechanism of new PB assembly was investigated as well as its possible implication in tumorigenesis in the case of colorectal cancer.

The manuscript raises interesting questions and presents an appreciable piece of work. However, their findings suffer from a number of shortcomings as well as many over-interpretations, as exemplified below.

Major concerns

General comments

(1) The material and methods part and figure legends are minimalist and methodological details should be added in order to facilitate the understanding of the experiments.

(2) The authors very often claim their findings while only referring to a figure number. Additional explanations would be useful.

Specific points

(3) In the CRISPR-Cas9 experiment, aiming in HAX1 knock out, the author gave very few technical information. In particular, we do not know how many cellular clones with invalidated HAX1 were used for the analyses presented in Fig. 5 and later, which is a concern due to the elevated risk of off-target events in CRISPR-editing experiments. This casts some doubt on the findings since the data shown could be the consequence of an additional invalidation of another gene. In the same vein, authors should test the expression level of 4E-T alongside with DDX6 and LSM14A (extended Fig. 5a), since the former protein is also essential to PB assembly (Ayache et al MBoC 2015, already cited by the authors).

(4) Display and comparisons of RNAseq dataset should be improved (Extended Fig. 5 d and 5e), notably using volcano plots

that allow a more precise comparison of 2 sets of data. In addition, the comparison of untreated WT and HAX1 KO cells is lacking (only results from oligomycin-treated cells are shown in Extended fig. 5d, 5e).

(5) The authors stated that mRNA decay occurs in PBs (p.6, l.8 and p.8, l.40). This has been controversial for years, but 2 recent studies using different approaches clearly showed that it is not the case in mammalian cells (Horvathova et al, Mol Cell 2017 and Hubstenberger et al, Mol Cell 2017, the latter being already cited by the authors). This erroneous statement may have biased the study (cf p.6, l.8-9).

(6) A cell fractionation has been performed aiming to obtain a P-body-enriched fraction. First, the manuscript does not display any technical detail. Second, referring to the cited papers, the fractionation consists of a 10000g centrifugation of a cytoplasmic extract. Thus, the pellet contains all cytoplasmic organelles of the cell, including its numerous membrane-less granules (and not only or mostly PBs). Therefore, the interpretations of the results presented in Fig. 5e-i are overstated (e.g. p.6 : "HAX1 is the pioneer protein during P-body formation", "the recruitment of LSM14A and DDX6 into P-bodies is dependent on the ubiquitination of HAX1").

(7) Concerning HAX1, all imaging data were obtained following overexpression of the protein. Given that the authors frequently mentioned a "TRIM23/HAX1/P-body pathway" in the manuscript and proposed that "HAX1 might be the key scaffold molecule of the P-body" (p.9, l.2), some data on the endogenous HAX1 would be helpful. Is endogenous HAX-1 concentrated in PBs in their cell lines? Does it co-localize with LSM14A, DDX6, 4E-T, DCP1a? Moreover, testing PB assembly following the knock-down of the endogenous HAX-1 would more convincingly indicate its requirement for this process. Especially as the group cited by the authors for the finding that HAX-1 is a PB component also showed in the same paper that the knock-down of this protein leads to an increase of PB number (Zayat et al, DNA Cell Biol 2015).

Minor concerns

(8) In Fig.1a, 1e, 1g, the PB size seems to be heterogeneous in treated conditions. This observation could be quantified.

(9) Fig. 5j, 5k : the immunoprecipitation data are hardly comparable due to differential level of immunoprecipitated proteins (endogenous HAX1 and Flag-HAX1).

(10) The morphology of stress granules presented in Extended Fig. 5b (small size and regular shape) is unusual. This may be specific to this cell type, but an additional stress granule marker (e.g. TIA1) should be tested in order to strengthen these first data.

(11) Electron transfer chain factors of OXPHO could be highlighted in Extended Fig. 5f, 5g and/or in the corresponding legend.

Referee #3:

This manuscript by Zhan and colleagues puts forward an interesting model in which the E3 ligase TRIM23 ubiquitinates HAX1 to promote HAX1-mediated assembly of P-bodies. They show some evidence that this occurs uniquely in response to energy stress (e.g., inhibition of the ETC). Strengths of the paper include orthogonal approaches to validate important conclusions. The manuscript covers a solid breadth of the mechanism and includes use of CRISPR screening to identify the E3 ligase (Trim23) involved in P-body formation during nutrient stress and its substrate (HAX1). They also extend the work to a clinical setting by implicating the cytoplasmic translocation of Trim23 as a potentially important biomarker in colorectal cancer. Although the manuscript as a whole is quite strong, it has a few key weaknesses that should be addressed before publication.

Major critiques:

The authors need control/validation experiments to ensure that the UAE inhibitors are working at the given concentration and timepoint. For example, an anti-ubiquitin immunoblot with and without the inhibitors would suffice.

The CRISPR screen experiment system needs a much more thorough explanation. It is not clearly explained how the CRISPR screen was set up and why cells with moderate LMS14A-GFP expression were initially selected for screening, but then cells with high LMS14A-GFP were selected for sgRNA sequencing. Also, wouldn't deletion of Trim23 reduce P-body formation/LMS14A-GFP fluorescence? This is a confusing part of the manuscript.

Related to the point above, more information is needed on the E3 ligases identified in the screen. The authors say that they chose Trim23 because Trim24 plays a role in p-body assembly. Was this the only reason? This seems fairly tenuous. Did the authors look at any other E3s? For example, if they looked at other E3s but the data were negative, this information is important and should be shared with the community. Also, shouldn't the CRISPR screen results be available for the research community (i.e., a table of E3 ligases identified in the screen)?

In figure 7, the conclusion that the lowest and high levels of HAX1 ubiquitination correlated with the best and worst survival seems weak given that the authors weren't actually looking at HAX1 ubiquitination. How do the authors justify high or low levels of HAX1 as a surrogate marker for HAX1 ubiquitination? That is not well explained in the text. For example, if ubiquitination of HAX1 didn't affect total proteins levels in previous experiments, why is high HAX1-content in the cells taken as a marker of ubiquitinated HAX1? Additionally, based on these observations, most of the overexpressed TRIM23 in previous experiments must have been mostly cytosolic to be able to ubiquitinate HAX1. Is this backed up by data?

Other conclusions that were a bit overstated:

- The authors assert that Trim23 exists in liquid droplets (e.g., page 5 lines 2-3), but no data are shown to really prove this is the case. Seems more accurate to simply say that they colocalize with markers of P-bodies
- The authors describe the E3 ligase Trim23 "responding" to nutrient stress. It's hard to say whether it's the enzyme responding, the substrate (e.g., by becoming more available), etc. I would re-state and tone down the conclusions in this regard.

Minor critiques:

Line 27 on page 5 says they made a "prediction" on phosphositeplus. How was the prediction made? Number of unique mass spec identifications? Is this the total number of reported Ub-lysines on the protein. More explanation is needed here to explain why these ub-lysines were chosen.

Page 11, lines 24-25. It would be helpful to provide more detail on how the S2 and P2 pellets were produced.

Response to reviewers' comments

Referee #1:

In this paper, Zhan et al. show that ATP depletion leads to an increase in the number of P-bodies per cell. They demonstrate that this increase in P-body number depends on the ubiquitination of HAX1 at K131 by the E3 ubiquitin ligase TRIM23. Furthermore, HAX1 ubiquitination, as well as functional TRIM23, are required for in vitro condensation of HAX1. Next, the authors show that HAX1 is an essential P-body component functioning early in P-body assembly. Finally, they present evidence that TRIM23 and HAX1 are involved in colorectal cancer proliferation.

The presented work contains an impressive amount of data, many of which are valuable additions to the field. However, to make the story more accessible we suggest that the authors consider to revise the manuscript and reduce the amount of data displayed in this work.

We appreciate the reviewer's positive and constructive comments on our study. To make the story more accessible, we removed the RNA-seq section (original **Fig EV5d-g**) and the CRC cell transwell data (original **Fig 7h-m**) from the revised manuscript. We also included additional figures to address each concern raised by this reviewer.

In addition, the work provides insight into a new aspect of P-body formation and draws a connection between P-bodies and energy stress. Yet, the connection between energy depletion and heat shock is not entirely clear, considering that ATP reduction is only one of many effects of heat shock response. Furthermore, the effects on ATP reduction are marginal compared to oligomycin treatment and rescue with ATPsome and NADH+CoQ10 increases the ATP levels by 1.5 and 2.5-fold respectively.

Thanks for pointing out this issue. We agree with the reviewer's concern about the connection between energy stress and heat shock. We deleted these related statements in the revised manuscript and just regarded heat shock as one of the stimulating factors that induces P-bodies formation. This modification had no effect on the main findings of our study.

The authors need to consider that this possibly results in hydrotropic effects of ATP on condensates dissolution.

This is indeed an important point. To directly explore this possibility, we examined the effect of ATP on HAX1 LLPS in vitro. Supplement of ATP had negligible impact on HAX1 droplets formation (**Appendix Fig S1B** and **Reviewer Figure 1A**), suggesting that the dissolution of HAX1 condensates is independent of the hydrotropic effects of ATP. In addition, neither the ATPsome nor the NADH+CoQ10 combination could reduce the number of P-bodies in *UBB*- or *TRIM23*-deficient cells (**Reviewer Figure 1B-E**), proving that this process is mediated by Ub and TRIM23 rather than by ATP itself.

Finally, the involvement of the axis HAX1/TRIM23 in colorectal cancer is well developed but the involvement of P-bodies is merely a correlation and the authors need to discuss it as such.

We agree with the reviewer that the involvement of P-bodies in CRC is not well established. We thus carried out several experiments to answer this question. First, we knocked out *LSM14A* in HCT116 cells by CRSIPR/Cas9 and found that P-bodies labeled with LSM14A and DCP1A almost disappeared (new **Appendix Fig S3A, B**). Overexpression of TRIM23 in LSM14A-knockout HCT116 cells did not induce P-bodies formation (new **Appendix Fig S3A, B**) and promote cell proliferation in vitro (**Fig 8N**). Second, reintroduction of HAX1 into LSM14A/HAX1 double knockout HCT116 cells also did not support the formation of P-bodies labeled with DDX6 and DCP1A, although HAX1 granules were normal (new **Appendix Fig S3C-E**). Reintroduction of HAX1 did not promote the proliferation of LSM14A/HAX1 double-knockout HCT116 cells (**Fig 8O**). Third, disruption of P-bodies with 1,6-hexanediol (1,6-HD), an aliphatic alcohol disassembling LLPS-dependent macromolecular condensate, inhibited the ability of TRIM23 and HAX1 to promote HCT116 cell proliferation (**Appendix Fig S3F-K**). Taken together, these data provide evidence that the function of TRIM23/HAX1 in the tumorigenicity of CRC depends on P-bodies.

In summary, this paper provides convincing evidence for a role of ubiquitination of HAX1 by TRIM23 for the formation of P-bodies. However, the results on P-body induction by energy depletion and the overall OXPHOS gene reduction in HAX1 KO and HAX1K131R are less convincing. Nevertheless, upon adequate revision the work should be published in EMBO Journal.

We appreciate the reviewer's positive comments on our study.

Major Points

- The authors repeatedly conclude that P-bodies are the responsible factors for the observed effects. This conclusion cannot be drawn with the presented evidence and need to be toned down and any implied causal connection between P-bodies and downstream effects should be revised and rephrased. For example, an importance of TRIM23/HAX1 for the tumorigenicity of CRC is supported, yet any importance of P-bodies for tumor development is not, and the author need to consider that TRIM23/HAX1 might be able to cause the observed effect independently of P-bodies. The same applies to the conclusions on phase separation. The conclusion in the title that "phase separation of HAX1 triggers P-body assembly" is based on the observation that HAX1 can phase separate in vitro and that HAX1_{K131R} as well as HAX1_{ΔIDR2} disrupts P-body assembly in cells. However, the K131 ubiquitination could also control P-body assembly in ways unrelated to phase separation. Therefore, the presented data should not be presented as depicting a causal relationship between phase separation of HAX1 and P-body formation.

We appreciate the insightful points that the reviewer has raised and have included extensive new data to address the raised concerns. As explained above (in response to the overall comment), we have conducted several experiments to verify the involvement of P-bodies in colorectal cancer (**Fig 8N, O** and **Appendix Fig S3**). To establish the relationship between the LLPS of HAX1 and P-bodies formation, we added the FUS N-terminal IDR region (IDR^{FUS}) to the HAX1^{K131R} and HAX1^{ΔIDR2} mutants to induce LLPS. The HAX1^{ΔIDR2}-IDR^{FUS} and HAX1^{K131R}-IDR^{FUS} fusion proteins formed round condensates in the cytoplasm, and P-bodies also re-formed (**Appendix Fig S1C-F**). These data established a causal relationship between the LLPS of HAX1 and P-bodies formation. Moreover, the surface sensing of translation (SUnSET) assay showed that HAX1^{K131R}-IDR^{FUS} or HAX1^{ΔIDR2}-IDR^{FUS} repressed overall protein synthesis, similar to WT HAX1 (new **Fig 7G, H**), suggesting that the downstream effects of P-bodies, namely, the inhibition of protein synthesis, were also regulated by LLPS of HAX1. As suggested by the reviewer, we also revised the manuscript to avoid overstating the conclusion derived from the presented data.

- The abstract should be carefully re-written in a logical manner avoiding unclear expressions like "unnecessary biological process restriction". It would be advisable that a native English speaker proofreads the article especially regarding logical consistency of the writing.

We have re-written the abstract and invited a native English speaker to proofread the manuscript.

- "We also found that heat shock decreased cellular ATP levels, which could be reversed by supplement with ATPsome or CoQ10+NADH (Extended Data Fig. 1f, g), suggesting that low ATP levels were responsible for P-Body assembly induced by heat shock." This statement is not supported by data. The reduction of ATP in HS condition is lower compared to the 2-DG treatment which alone barely increases P-Body number. While HS reduces ATP levels it might have additional effect on PB formation independent of changes in ATP level. The authors need to comment and discuss this. Furthermore, if low ATP levels are responsible for the increase in P-body numbers, then colorectal cancer cell lines should display no P-bodies if treated with ATPsome or CoQ10+NADH but increase their number after oligomycin treatment. If this is not the case the initial general statement should be toned down as "Low ATP levels increases P-bodies number in a cell line dependent context"

We agree with the reviewer's comment that the statement "low ATP levels were responsible for P-body assembly induced by heat shock" was not supported by data and have deleted this conclusion in the revised manuscript. We have shown that oligomycin treatment increased the number of P-bodies in HCT116 cells (**Fig EV2F, G**) and included new data showing that RKO cells also displayed increasing P-bodies when treated with oligomycin (**Fig EV1E, F**) but decreased their number after CoQ10+NADH treatment (**Fig EV1H, I**). ATPsome slightly reduced the number of P-bodies under basal conditions but markedly prevented oligomycin-induced P-bodies formation in RKO cells (new **Fig EV1M, N**). Thus, low ATP levels also increase the number of P-bodies in CRC cell lines.

- It is not clear why sometimes 2-DG+Oligomycin is used and at other times only oligomycin considering they have a different effect on reducing ATP levels. Could the author comment on this and/or clarify this in the text?

2-DG plus Oligomycin was used at the beginning to produce different ATP levels (**Fig 1A, B and Fig EV. 2A**). We chose oligomycin for the subsequent experiments because oligomycin alone was sufficient to increase the number of P-bodies from one to four per HEK293T cell, a response comparable to that induced by the combination of 2-DG plus oligomycin. The only exception was the original **Fig. 1g**, in which 2-DG+Oligomycin was selected to prove that ATPsome could reverse the phenotype induced by the most reduced ATP. We also showed similar results for oligomycin alone in the original **Extended Data Fig. 1h**. We have included this figure as the new **Fig 1G, H** and added the sentence "We chose oligomycin alone for ATP depletion in subsequent experiments" to the revised manuscript.

- "Because the core function of P-body is mRNA degradation, we focused on oligomycin-repressed genes". This statement is not correct as a large body of literature shows that P-bodies are not sites of mRNA decay. Furthermore, Extended Figure 5d-g is difficult to interpret and should be better described/analyzed in the text. Especially, the oligomycin treated vs untreated samples need to be clarified the corresponding fraction of genes belonging to the different groups have to be made clear. On the same line "The most significantly downregulated genes in wild-type cells were electron transfer chain genes of OXPHO, which could not be repressed by oligomycin in HAX1 knockout and K131R mutant cells (Extended Data Fig. 5f, g)". Carefully looking at the graph and comparing the oligomycin treated lines, it seems that the mitochondrial genes are similarly repressed in HAX1 WT, HAX1 KO and HAX1K131R. Can the authors provide side by side quantifications of these genes for better comparison?

Thank you for pointing out this issue. We re-analyzed the RNA-seq data according to the reviewer's suggestion and displayed the results as volcano plot and heatmap (**Reviewer Figure 2A-E**). The upregulated and downregulated genes after oligomycin treatment were divided into three categories: genes whose expression changed only in WT cells (WT only), genes whose expression changed both in WT and HAX1 KO/HAX1^{K131R} cells (WT \cap HAX1 KO/HAX1^{K131R}), and genes whose expression changed only in HAX1 KO/HAX1^{K131R} cells (HAX1 KO/HAX1^{K131R} only). The electron transfer chain genes of OXPHO were labeled, and the results showed that 13 genes were downregulated after oligomycin treatment in wild-type cells but that only 4 genes (ND1, ND2, ND5, and ND6) were downregulated in HAX1 KO cells and that 5 genes (ND1, COX1, ND5, ND6, and COX3) were downregulated in HAX1^{K131R} cells (**Reviewer Figure 2D, E**).

These genes were further quantified by FPKM (fragments per kilobase per million mapped fragments) as suggested (**Reviewer Figure 2F**). Only ATP6/8 and ND4 were not downregulated significantly in HAX1 knockout or K131R mutant cells after oligomycin treatment, while ND1/2/5/6 were decreased in all cells. For COX2, CYTB, ND3 and ND4L, although the fold change was lower in HAX1 knockout and K131R mutant cells than in WT cells, the difference was still statistically significant. Thus, the conclusion that OXPHO genes could not be repressed in HAX1 KO and HAX1K131R cells was indeed inaccurate.

As the reviewer mentioned, P-bodies are not sites of mRNA decay, and we believe that changes in the transcriptome might not be suitable for characterizing PB function. Thus, we decided to remove the RNA-seq data from the revised manuscript.

To explore the function of TRIM23/HAX1 in mammalian cells, we assessed the effect of TRIM23/HAX1 on protein synthesis via the surface sensing of translation (SUnSET) assay. We found that TRIM23 suppressed overall protein synthesis in HEK293T and CRC cells (new **Fig 7A-D**). Similarly, knockout of HAX1 promoted overall protein synthesis (new **Fig 7E**), while overexpressing WT HAX1, but not K131R mutant HAX1, suppressed protein synthesis (new **Fig 7F**). Oligomycin treatment suppressed protein synthesis in WT cells, but this effect was absent in TRIM23 KO and HAX1 KO cells (new **Fig 7C-F**). Moreover, the addition of IDR^{FUS} made HAX1 Δ IDR2 and HAX1^{K131R} regain the ability to inhibit protein synthesis (new **Fig 7G, H**). These results suggested that TRIM23/HAX1 was essential for translational inhibition induced by energy stress, which was mediated by P-bodies.

- Several references are not correct and do not refer to the points made. For example, references 42,43 do not support the following claim: "Many P-body proteins possessing LLPS ability harbor IDRs, which are considered in the context of the regulation of P-body assembly". The authors need to check all references carefully.

We have checked all references carefully, and changed the original references 42,43.

Minor points

- DCP1 α should be changed in DCP1A

Fixed.

- NaAsO₂ should be changed to sodium arsenite to be consistent with figures.

Fixed.

- μ M has to be changed to μ M

Fixed.

- One consistent form of writing "P-bodies" should be used

Fixed.

- Page 9 line 9: change aerobic with anaerobic

Fixed.

- The axis label "Time(s)" should be " Time (s)" to distinguish from the plural

Fixed.

- For those experiments where only technical replicates are used, it needs to be explained why.

We apologize for the mistakes here. It should be biological replicates and we have corrected this in the revised manuscript.

- Can the authors comment on the observation that an increase in ATP levels does not decrease P-body numbers in steady state levels but only under heat shock?

We claimed that an increase in ATP levels slightly reduced the number of P-bodies in steady state levels and markedly prevented P-bodies formation under heat shock. This is likely because ATP levels regulate P-bodies formation through ubiquitination, and protein ubiquitination is relatively low in steady state and higher under heat shock. As shown in **Reviewer Figure 3**, compared with in steady state, NADH+CoQ10 or ATPsome treatment led to a more significant reduction of protein ubiquitination under heat shock.

- In Figure 2b the addition of ATPsome seems to decrease the ubiquitin levels also without the addition of oligomycin. Can the authors comment on this and discuss how this might affect their conclusions?

There is a possibility that low ATP level might promote ubiquitination not only by activating E3 ligases (e.g., TRIM23) but also by suppressing the activity of deubiquitinase. If so, the addition of ATPsome could enhance deubiquitination under both basal and stress conditions. We indeed observed that ATPsome treatment reduced the ubiquitin levels induced by heat shock (**Reviewer Figure 3**). This speculation is also consistent with the finding that ATPsome treatment slightly reduced the number of P-bodies in steady state levels.

- Figure 5J does not include any blot for DCP1A, yet in the text it is specified that there is no binding.

We are sorry for missing display of DCP1A. We have included the blot for DCP1A in the revised manuscript (**Fig 5J**).

- Extended data figure 1F: The control should be DMF not DMSO

We have revised the label (**Fig EV1G**).

- Extended data figure 1J: Statistics should be checked

We checked the statistics in the original **Extended data figure 1J** but did not find any abnormalities. We deleted the original **Extended data figure 1J** in the revised manuscript because its associated figure (original **Fig. 5G**) has been removed as we noted above.

- Time points for drug treatment and heat shock are missing. This information needs to be provided at least in the methods.

The time points for drug treatment and heat shock have been included in the figure legends and methods.

- Figure panels should be rearranged to fit the logical sequence of the text wherever possible.

We have rearranged all the Figure panels.

- Given the many different cell lines used to the authors should label each figure with the corresponding cell line for better understanding.

We have labeled the cell lines below every figure.

- The statement "In contrast, forced ubiquitin expression facilitated the recovery of LSM14A after FRAP, suggesting that ubiquitination is necessary in P-body LLPS (Extended Data Fig.2m, n)" is not a valid conclusion because a faster FRAP recovery is not necessarily connected to LLPS.

We agree with this comment and have replaced "LLPS" with "dynamics" in the revised manuscript.

- Please refrain from using green and red in a merged image.

Yellow and pink were used instead of green and red in the merged images.

- Can the authors comment on their claim that "PYR-41 treatment also significantly shortened the recovery time of LSM14A (Fig. 2k, l)". It looks rather that the mobile fraction is reduced.

We apologize for the incorrect description here. We corrected the statement to "with PYR-41 treatment significantly slowing down the FRAP process" in the revised manuscript.

- Figure E4C "anti-flag" is spelled wrong.

We have corrected this spelling error (new **Fig EV4D**).

- "Taken together, our findings validate that ubiquitinated HAX1 is required for cells to respond to energy stress by controlling P-body assembly." Can the authors elaborate what "response" they refer to?

The response refers to the inhibition of protein synthesis mediated by P-bodies.

- Claiming that HAX1 is the pioneer protein for P-body assembly is premature because not all P-body proteins have been tested. You can say that is "a pioneer protein" though.

As suggested, we replaced "the pioneer protein" with "a pioneer protein" in the revised manuscript.

Referee #2:

In this manuscript, Zhan et al studied P-body (PB) formation and dynamics in the context of the energy stress consecutive to ATP depletion. By imaging stressed human cells, including FRAP experiments, they showed that ATP depletion induced de novo PB assembly in an ubiquitin dependant manner. By using a CRISPR screen and various knock-out, knock-down and over-expression experiments, they found that the E3 ligase TRIM23 was involved in response to ATP stress through HAX1 ubiquitination. The mechanism of new PB assembly was investigated as well as its possible implication in tumorigenesis in the case of colorectal cancer.

The manuscript raises interesting questions and presents an appreciable piece of work. However, their findings suffer from a number of shortcomings as well as many over-interpretations, as exemplified below.

We thank the reviewer for his/her interest in our work. We have included new data to overcome the shortcomings and revised the manuscript to avoid over-interpretations.

Major concerns

General comments

(1) The material and methods part and figure legends are minimalist and methodological details should be added in order to facilitate the understanding of the experiments.

More detailed methods and figure legends were included in the revised manuscript.

(2) The authors very often claim their findings while only referring to a figure number. Additional explanations would be useful.

We have included additional explanations for the figures as suggested by the reviewer.

Specific points

(3) In the CRISPR-Cas9 experiment, aiming in HAX1 knock out, the author gave very few technical information. In particular, we do not know how many cellular clones with invalidated HAX1 were used for the analyses presented in Fig. 5 and later, which is a concern due to the elevated risk of off-target events in CRISPR-editing experiments. This casts some doubt on the findings since the data shown could be the consequence of an additional invalidation of another gene. In the same vein, authors should test the expression level of 4E-T alongside with DDX6 and LSM14A (extended Fig. 5a), since the former protein is also essential to PB assembly (Ayache et al MBoC 2015, already cited by the authors).

We apologize for missing the description regarding HAX1 knockout by CRISPR-Cas9. We added detailed technical information to the section "CRISPR Mediated Knockout Cells" of the "MATERIALS AND METHODS". The HAX1 knockout HEK293T cells were monoclonal, and the HAX1 knockout HCT116 cells were polyclonal.

Initially, we knocked out HAX1 with two individual sgRNAs in HEK293T cells and found that both the average number of P-bodies in per cell and the percentage of cells with P-bodies were reduced in the unsorted cells (**Reviewer Figure 4**). We also analyzed the P-bodies in the cells with HAX1 knockdown, and the results were consistent with the HAX1 KO cells (**Fig EV5A-C**). Combined with the finding that re-expressing WT HAX1 could rescue P-bodies in the HAX1 KO cells, the probability of off-target effects inducing the phenotype was very low.

We have added 4E-T detection by immunoblotting as suggested by the reviewer and found that the expression level of 4E-T did not change in HAX1 KO cells (**Fig EV5D**).

(4) Display and comparisons of RNAseq dataset should be improved (Extended Fig. 5 d and 5e), notably using volcano plots that allow a more precise comparison of 2 sets of data. In addition, the comparison of untreated WT and HAX1 KO cells is lacking (only results from oligomycin-treated cells are shown in Extended fig. 5d, 5e).

Sorry for the confusing presentation. The "WT" in the original **Extended Fig. 5d** represents the upregulated genes in oligomycin-treated WT cells compared with the untreated WT cells. We improved the display and comparisons of the RNA-seq dataset according to the reviewer's suggestion (**Reviewer Figure 2**). Since changes in the transcriptome might not be suitable for characterizing PBs' function, we thus decided to remove the RNA-seq data from the revised manuscript.

(5) The authors stated that mRNA decay occurs in PBs (p.6, l.8 and p.8, l.40). This has been controversial for years, but 2 recent studies using different approaches clearly showed that it is not the case in mammalian cells (Horvathova et al, Mol Cell 2017 and Hubstenberger et al, Mol Cell 2017, the latter being already cited by the authors). This erroneous statement may have biased the study (cf p.6, l.8-9).

We thank the reviewer for this important point. We agree that mRNA decay did not occur in PBs and that the change in the transcriptome according to the RNA-seq data was not suitable here. We have deleted the RNA-seq data and revised the related statement in the revised manuscript. Instead, we further assessed the effect of TRIM23/HAX1 on protein synthesis by the surface sensing of translation (SUnSET) assay and found that TRIM23/HAX1 was essential for translational inhibition induced by energy stress (new **Fig 7**).

(6) A cell fractionation has been performed aiming to obtain a P-body-enriched fraction. First, the manuscript does not display any technical detail. Second, referring to the cited papers, the fractionation consists of a 10000g centrifugation of a cytoplasmic extract. Thus, the pellet contains all cytoplasmic organelles of the cell, including its numerous membrane-less granules (and not only or mostly PBs). Therefore, the interpretations of the results presented in Fig. 5e-i are overstated (e.g. p.6 : "HAX1 is the pioneer protein during P-body formation", "the recruitment of LSM14A and DDX6 into P-bodies is dependent on the ubiquitination of HAX1").

Thanks for pointing out this issue. We have included additional details of the fractionation assay in the revised manuscript. The fractionation assay by differential centrifugation was not used to obtain purified P-bodies but rather to distinguish the P-bodies enriched insoluble fraction from the soluble cytoplasm. Although the P2 pellet contained only partially purified P-bodies, this assay could be applied for the semiquantification of protein condensation and distribution. As mentioned by the reviewer, these results are indeed not the most direct evidence supporting the translocation of proteins (HAX1, LSM14A and DDX6) into P-bodies. Thus, we revised the description and conclusion to avoid overstatement in this section (e.g. "the condensation of LSM14A and DDX6, which is the key process of P-bodies formation, is dependent on the ubiquitination of HAX1").

(7) Concerning HAX1, all imaging data were obtained following overexpression of the protein. Given that the authors frequently mentioned a "TRIM23/HAX1/P-body pathway" in the manuscript and proposed that "HAX1 might be the key scaffold molecule of the P-body" (p.9, l.2), some data on the endogenous HAX1 would be helpful. Is endogenous HAX-1 concentrated in PBs in their cell lines? Does it co-localize with LSM14A, DDX6, 4E-T, DCP1a? Moreover, testing PB assembly following the knock-down of the endogenous HAX-1 would more convincingly indicate its requirement for this process. Especially as the group cited by the authors for the finding that HAX-1 is a PB component also showed in the same paper that the knock-down of this protein leads to an increase of PB number (Zayat et al, DNA Cell Biol 2015).

We thank the reviewer for this important point. As suggested, we included additional data on endogenous HAX1. We found that endogenous HAX-1 was concentrated in PBs and co-localized with LSM14A, DDX6, 4E-T, and DCP1A in both HEK293T and HCT116 cells (**Fig EV4B, C**). We knocked down HAX1 in HEK293T cells and found that knocking down endogenous HAX-1 decreased the number of PBs (**Fig EV5A-C**). Moreover, we monitored the condensation of P-bodies core proteins (DCP1A, LSM14A and DDX6) during P-bodies formation. Consistent with the results in HAX1 KO cells, HAX1 knockdown suppressed the condensation of LSM14A and DDX6 but not that of DCP1A when TRIM23 was overexpressed or when energy stress was present (**Fig EV5I, J**).

Minor concerns

(8) In Fig.1a, 1e, 1g, the PB size seems to be heterogeneous in treated conditions. This observation could be quantified.

Following the reviewer's suggestion, we counted PBs of different sizes in **Fig EV1D** (for Fig. 1A), **Fig EV1K** (for Fig.1E) and **Fig EV1L** (for Fig.1G). By measuring the diameter of PBs under different conditions via ImageJ software, we found that the median diameter was 0.2 μ M, which was used as a standard for distinguishing large from small PBs.

(9) Fig. 5j, 5k: the immunoprecipitation data are hardly comparable due to differential level of immunoprecipitated proteins (endogenous HAX1 and Flag-HAX1).

To make the immunoprecipitation data comparable, we performed immunoprecipitation with an anti-HAX1 antibody to pull down the WT and K31R HAX1 proteins (new **Fig 5K**). We also transfected Flag-HAX1 into *HAX1* KO HEK293T cells and immunoprecipitated Flag-HAX1 with a Flag antibody (**Reviewer Figure 5**).

(10) The morphology of stress granules presented in Extended Fig. 5b (small size and regular shape) is unusual. This may be specific to this cell type, but an additional stress granule marker (e.g. TIA1) should be tested in order to strengthen these first data.

Thanks for your suggestion. We labeled stress granules with TIA1 and G3BP2, as shown in **Fig. EV5B**, and found that HAX1 knockout did not affect stress granules formation under either basal or sodium arsenite conditions (**Fig EV5E**). We also labeled stress granules with TIA1 and G3BP2 in **Fig EV1B** and **Fig EV3P**.

(11) Electron transfer chain factors of OXPHO could be highlighted in Extended Fig. 5f, 5g and/or in the corresponding legend.

We labeled the electron transfer chain factors of OXPHO according to the reviewer's instructions (**Reviewer Figure 2**). Yet, we decided to remove the RNA data from the revised manuscript as explained above,

Referee #3:

This manuscript by Zhan and colleagues puts forward an interesting model in which the E3 ligase TRIM23 ubiquitinates HAX1 to promote HAX1-mediated assembly of P-bodies. They show some evidence that this occurs uniquely in response to energy stress (e.g., inhibition of the ETC). Strengths of the paper include orthogonal approaches to validate important conclusions. The manuscript covers a solid breadth of the mechanism and includes use of CRISPR screening to identify the E3 ligase (Trim23) involved in P-body formation during nutrient stress and its substrate (HAX1). They also extend the work

to a clinical setting by implicating the cytoplasmic translocation of Trim23 as a potentially important biomarker in colorectal cancer. Although the manuscript as a whole is quite strong, it has a few key weaknesses that should be addressed before publication.

We appreciate the positive comments from this reviewer on our study.

Major critiques:

The authors need control/validation experiments to ensure that the UAE inhibitors are working at the given concentration and timepoint. For example, an anti-ubiquitin immunoblot with and without the inhibitors would suffice.

We have added validation experiments showing that TAK-243 and PYR-41 inhibited poly-ubiquitylation in HEK293T cells (**Fig. EV2B**) and HCT116 cells (**Fig. EV2E**) in the revised manuscript.

The CRISPR screen experiment system needs a much more thorough explanation. It is not clearly explained how the CRISPR screen was set up and why cells with moderate LMS14A-GFP expression were initially selected for screening, but then cells with high LMS14A-GFP were selected for sgRNA sequencing. Also, wouldn't deletion of Trim23 reduce P-body formation/LMS14A-GFP fluorescence? This is a confusing part of the manuscript.

We apologize that we did not describe the CRISPR screen experiment clearly. CRISPR screening was more sensitive in homogeneous cells. Considering that the numbers of P-bodies in individual LSM14A-GFP-expressing HEK293T cells were heterogeneous, cells with moderate LMS14A-GFP expression were initially selected for screening. The CRISPR screening was based on the finding that the fluorescence intensity of LSM14A-GFP detected by flow cytometry was positively correlated with the number of P-bodies (**Fig. EV3A-C**). Thus, we could isolate the cells with more P-bodies from the cell population by fluorescence-activated cell sorting. When sgRNAs targeting the positive regulatory factors of P-bodies were introduced into cells, the fluorescence intensity of LSM14A-GFP was weakened and the percentage of cells containing these sgRNAs would be reduced in populations with high LSM14A-GFP. These sgRNAs would be lost in the cells with high LMS14A-GFP expression. We have included additional explanation in the revised manuscript.

It was true that deletion of TRIM23 reduced P-body formation/LMS14A-GFP fluorescence. Thus, in the screening assay, the percentage of TRIM23 KO cells was lower and the sgRNAs against *TRIM23* were depleted in cells with high LMS14A-GFP expression.

Related to the point above, more information is needed on the E3 ligases identified in the screen. The authors say that they chose Trim23 because Trim24 plays a role in p-body assembly. Was this the only reason? This seems fairly tenuous. Did the authors look at any other E3s? For example, if they looked at other E3s but the data were negative, this information is important and should be shared with the community. Also, shouldn't the CRISPR screen results be available for the research community (i.e., a table of E3 ligases identified in the screen)?

We tested ten of the top-ranking positive regulators using individual CRISPR-based knockouts. Deletion of *LNX1*, *UBE3* and *TRIM23* reduced the number of P-bodies (**Fig EV3D, E**). We selected TRIM23 for further study in this work, which was partially inspired by TRIM24 and because of its powerful effect on P-bodies. We have uploaded the CRISPR screen results (**Table EV1**) in the revised version.

In figure 7, the conclusion that the lowest and high levels of HAX1 ubiquitination correlated with the best

and worst survival seems weak given that the authors weren't actually looking at HAX1 ubiquitination. How do the authors justify high or low levels of HAX1 as a surrogate marker for HAX1 ubiquitination? That is not well explained in the text. For example, if ubiquitination of HAX1 didn't affect total protein levels in previous experiments, why is high HAX1-content in the cells taken as a marker of ubiquitinated HAX1? Additionally, based on these observations, most of the overexpressed TRIM23 in previous experiments must have been mostly cytosolic to be able to ubiquitinate HAX1. Is this backed up by data? We are very sorry to confuse you here. As stated in the text, "Due to the lack of a specific antibody for ubiquitinated HAX1, we defined the level of ubiquitinated HAX1 by the localization of TRIM23 and the protein levels of HAX1..... ", we considered high levels of HAX1 along with the cytoplasmic location of TRIM23 as the surrogate marker for high HAX1 ubiquitination and low levels of HAX1 along with the nuclear location of TRIM23 as low HAX1 ubiquitination. We have added more explanation into the revised manuscript to clarify this point.

When we overexpressed GFP-TRIM23 in HEK293T cells, most of the GFP-TRIM23 was located in the cytoplasm (**Fig 4C**). Similarly, we found that overexpressed TRIM23 in HCT116 cells was mostly localized in the cytosol, as determined by nuclear and cytoplasmic fractionation assay (**Reviewer Figure 6A**). Moreover, cytoplasmic TRIM23 promoted more HAX1 ubiquitination than did TRIM23-NLS, which was located in both cytoplasm and nucleus (**Reviewer Figure 6B**). Hence, most of the overexpressed TRIM23 in these experiments were mostly cytosolic and were able to ubiquitinate HAX1.

Other conclusions that were a bit overstated:

- The authors assert that Trim23 exists in liquid droplets (e.g., page 5 lines 2-3), but no data are shown to really prove this is the case. Seems more accurate to simply say that they colocalize with markers of P-bodies

We have modified the statement as suggested by the reviewer.

- The authors describe the E3 ligase Trim23 "responding" to nutrient stress. It's hard to say whether it's the enzyme responding, the substrate (e.g., by becoming more available), etc. I would re-state and tone down the conclusions in this regard.

We have revised the manuscript to avoid overstatement.

Minor critiques:

Line 27 on page 5 says they made a "prediction" on phosphositeplus. How was the prediction made? Number of unique mass spec identifications? Is this the total number of reported Ub-lysines on the protein. More explanation is needed here to explain why these ub-lysines were chosen.

PhosphoSitePlus provides information of protein post-translational modifications (PTMs), including ubiquitylation, which was obtained from reported mass spectrometry (MS) data. The term "prediction" is inappropriate here, and we have changed the statement to "Five ub-lysines in HAX1 were identified by mass spectrometry (MS) according to the databases of post-translational modifications (**Fig EV4G**)".

Page 11, lines 24-25. It would be helpful to provide more detail on how the S2 and P2 pellets were produced.

We have added more detail on how the S2 and P2 pellets were produced in the methods section.

Dear Dr. Wang,

Thank you for submitting a revised version of your manuscript. Your study has now been seen by all original referees, who find that their previous concerns have been addressed and now recommend publication of the manuscript.

There remain only a few mainly editorial points that have to be addressed before I can extend formal acceptance of the manuscript:

- Please change the section order:

Title page - Abstract & Keywords - Introduction - Results - Discussion - Materials & Methods - Data Availability - Acknowledgments - Disclosure Statement & Competing Interests - References - Figure Legends - Tables with legends - Expanded View Figure Legends.

- Please resolve name discrepancies between the manuscript and our online system (Yongfeng Liu in the ms vs. Liu Yongfeng in eJP)

- Please use institutional e-mail addresses.

- Please provide ORCID IDs.

- Please enter your funding information into our online submission system (EJP)

- Please enter up to 5 keywords.

- Please remove the AC/CRedit section

- DATASET EV LEGENDS: Table EV1 looks like a Dataset and should be renamed and uploaded as such; the callout in the ms needs updating too; the correct nomenclature should be Dataset EV1

- Please provide the three Appendix Figures as a single PDF file (Appendix) that has a ToC with page numbers on the title page.

- Please provide source data as requested 30th Nov 23; For more information, please consult our website and the blank checklist which is uploaded.

- Please provide the synopsis image in jpeg, TIFF or png format and sized 550 pixels wide x 200-400 pixels high

- Please also provide a SYNOPSIS TEXT which consists of A) a short (1-2 sentences) summary of the findings and their significance, B) 3-4 bullet points highlighting key results

- Please check figure EV3 F and H for a possible re-use. SG Trim 23. Cells are the same.

- Figure Legends (main + EV): "1. Please define the annotated p values ****/* in the legend of figure 8c, e; as appropriate.

2. Please indicate the statistical test used for data analysis in the legends of figures 3b; 8c, e."

"1. Although 'n' is provided, please describe the nature of entity for 'n' in the legends of figures 1j; 6l-o.

2. Please note that the error bar is not defined in the legend of figure 5e.

3. Please note that the measure of center for the error bar needs to be defined in the legend of figure EV 1f."

"1. Please note that the white arrows are not defined in the legend of figure 1i; 2k, m; 3j, l; EV 2o; This needs to be rectified.

2. Please note that the red arrow is not defined in the legend of figure EV 4a. This needs to be rectified.

3. Please note that the letters a,b are not defined in the legend of figure 6c. This needs to be rectified."

With best regards,

Cornelius Schneider

Cornelius Schneider, PhD

Editor

The EMBO Journal

c.schneider@embojournal.org

When assembling figures, please refer to our figure preparation guideline in order to ensure proper formatting and readability in

print as well as on screen:

We realize that it is difficult to revise to a specific deadline. In the interest of protecting the conceptual advance provided by the work, we recommend a revision within 3 months (16th Jun 2024). Please discuss the revision progress ahead of this time with the editor if you require more time to complete the revisions. Use the link below to submit your revision:

Referee #1:

The authors have done an excellent job during the revision and have addressed all our major concerns.

Referee #2:

The revised version of the manuscript by Zhan and colleagues was substantially improved. The authors have addressed my concerns adequately. New data have been provided and the text now includes requested explanations and appropriate claims. I would support the publication of the revised manuscript.

Referee #3:

The authors satisfactorily addressed all of my concerns

All editorial and formatting issues were resolved by the authors.

Dear Prof. Wang,

I am pleased to inform you that your manuscript has been accepted for publication in the EMBO Journal.

Yours sincerely,

Cornelius Schneider, PhD
Editor
The EMBO Journal
c.schneider@embojournal.org
